# Scaffolding cooperation in human groups with deep reinforcement learning

**Kevin R. McKee** [1] ✉, **Andrea Tacchetti**[1], **Michiel A. Bakker**[1], **Jan Balaguer**[1], **Lucy Campbell-Gillingham**[1], **Richard Everett**[1] **& Matthew Botvinick**[1,2]

Effective approaches to encouraging group cooperation are still an open challenge. Here we apply recent advances in deep learning to structure networks of human participants playing a group cooperation game. We leverage deep reinforcement learning and simulation methods to train a 'social planner' capable of making recommendations to create or break connections between group members. The strategy that it develops succeeds at encouraging pro-sociality in networks of human participants ($N$ = 208 participants in 13 groups) playing for real monetary stakes. Under the social planner, groups finished the game with an average cooperation rate of 77.7%, compared with 42.8% in static networks ($N$ = 176 in 11 groups). In contrast to prior strategies that separate defectors from cooperators (tested here with $N$ = 384 in 24 groups), the social planner learns to take a conciliatory approach to defectors, encouraging them to act pro-socially by moving them to small highly cooperative neighbourhoods.

Cooperation is contagious. Social contact and interaction can spread pro-sociality from one person to another[1–3]. This property can cause cascades of cooperation in community settings, catalysing the accumulation of amity within groups and networks[4,5]. However, antisocial behaviour is also contagious[6]. Social networks thus have a corresponding tendency to propagate selfishness and other negative phenomena[7,8]. Such contagion dynamics pervade both personal social networks and contemporary social media[9,10], where an increasing amount of interpersonal interactions unfold[11–13]. Social planners face a challenge: how can one structure a community to scaffold and support cooperation, while mitigating the risk that defection will take hold?

Assortative mixing—a network phenomenon in which cooperators connect preferentially with other cooperators, and defectors with other defectors—is central to many prior solutions. For example, Rand, Arbesman and Christakis[14] provided individuals with random opportunities to make or break links to other community members, showing that link updates cause clustering among individuals sharing the same strategy and mitigate the natural decline in group cooperation. Similarly, Shirado and Christakis[15] embedded cooperative 'bots' throughout networks to foster homophilic clusters and promote cooperation. This line of research contends that assortative mixing prevents

antisocial contagion from corrupting altruistic behaviour by partitioning cooperators from defectors. It also frames assortment mechanisms in terms of punishment or ostracism: specifically, assortment threatens defectors with exclusion from the benefits of cooperative relationships[14–17]. In combination, these effects are believed to protect existing cooperators and punish defectors to incentivize changes to their behaviour. Indeed, studies of modern hunter–gatherer tribes indicate that cooperative assortment may trace back to early epochs of human evolutionary history[18,19].

Several recent research efforts propose using machine learning to identify novel solutions to social challenges (for example, refs. 20,21). Artificial intelligence (AI) and machine learning systems increasingly suffuse everyday social processes[22], so it seems natural to ask how they might support beneficial outcomes for human communities. For network-based problems, the application of machine learning is especially fitting: algorithms play a key role mediating the structure of online social networks[23–25]. Algorithms make recommendations to connect users, thus changing the structure of the underlying social graph.

In this Article, bringing these lines of research together, we aim to construct a social planner with deep learning that maximizes cooperation among human participants in a network cooperation game

[1]Google DeepMind, London, UK. [2]Gatsby Computational Neuroscience Unit, University College London, London, UK. ✉e-mail: kevinrmckee@google.com

(Fig. 1a; refs. 14,15; see also Supplementary Information Section A). Players are positioned on the vertices of a graph; edges represent active interpersonal links between players (Supplementary Fig. 1). Players accumulate (or lose) capital through turn-based interactions with their neighbours. On each turn, players choose to cooperate or defect. Cooperation exacts a constant cost $c = 0.05$ per linked neighbour from a player's capital. Each neighbour receives a constant benefit $b = 0.1$, generating net benefits for the neighbourhood at personal cost to the cooperator. Thus, group welfare is highest when everyone cooperates, but for each group member it is tempting to free-ride on the pro-sociality of others. Every turn, the social planner observes the graph structure and the players' most recent decisions (that is, their choice to cooperate or defect in the previous round). The planner then makes recommendations to the players as to which edges should be established or broken. Players decide whether to accept or reject the recommendations, resulting in changes to the graph connectivity. Subsequently, another turn begins. The game imposes no constraints on graph structure aside from precluding self-loops: with the right circumstances and recommendations, a social planner can produce outcomes as extreme as network isolates or fully connected graphs.

Here we leverage deep reinforcement learning and simulation methods to develop a new social planner capable of scaffolding cooperation among groups of interacting humans. The deep neural network tunes its parameters through repeated simulations of the cooperation game. Through this 'training' stage, the network refines its 'policy': a mapping from the state of the game (for example, the connectivity between players and their recent choices) to a probability distribution over actions for the planner to take (for example, recommendations to make to players). The policy starts out as a random mapping at the beginning of training, with the planner making random recommendations to players. Through reinforcement learning—and in particular, optimization through trial and error in simulation—the policy iteratively improves until the social planner is able to maintain cooperation at high levels in games with real human participants (the 'evaluation' stage). Neural networks can learn through interaction with real human groups, but the amount of trial-and-error experience needed for deep reinforcement learning takes a generally prohibitive amount of time to accumulate. Interactions with simulated human groups enable our social planning agent to gain a large amount of experience in a short period of time.

In specific terms, we construct a reinforcement learning agent with a graph neural network (a 'GraphNet'[26]). GraphNets explicitly encode graph structure into their computations (Fig. 1b,c). On a given turn of the network cooperation game, the GraphNet computes policy logits (representing a probability distribution over possible actions to take) and a value estimate (representing a prediction of future reward, given the current state of the game). Our reinforcement learning agent uses advantage actor–critic[27] as its learning algorithm.

The GraphNet-based agent trains to make rewiring recommendations by repeatedly playing as the social planner in simulation. Through games with simulated human players, the agent learns to effectively scaffold group cooperation. Across different random initializations of its neural network, the agent reliably converges to a high level of performance by the end of training (Supplementary Fig. 5). We select one of these high-performing agents to evaluate in 16-player games with human participants (the 'GraphNet social planner' condition; $N = 208$ participants across 13 groups).

To better contextualize the capabilities and behaviour of the GraphNet social planner, we compare its performance against several baseline strategies:

- In the 'static network' condition, the social planner never recommends any changes to the graph ($N = 176$ participants across 11 groups).

- In the 'random recommendations' condition, on each turn the social planner randomly samples 30% of the graph's possible edges and recommends that they be changed, creating edges if they are not active and breaking edges if they are already established (ref. 14; $N = 208$ participants across 13 groups).
- Finally, in the 'cooperative clustering' condition, the social planner uses a rule-based system to cluster cooperators ($N = 176$ participants across 11 groups). On each turn, the cooperative-clustering social planner makes recommendations that first disengage defectors from cooperators, and secondarily that connect cooperators with other cooperators[15]. Following prior implementation, the cooperative-clustering planner selects an additional 5% of the graph's possible edges at random and recommends that they be changed.

Across the GraphNet planner condition and all baseline conditions, we recruit $N = 768$ participants in 48 groups. Each group consists of 16 participants playing 15 rounds of the cooperative network game for real monetary stakes.

## Results

Following past studies[14–16,28], we employ generalized linear mixed models to analyse cooperation decisions at the individual level, with random effects for participants nested in groups. To evaluate all other group outcomes, we make use of group-level linear models. For both sets of models, visual inspections of residual and quantile–quantile plots suggest no practical issues with assuming normality and equal variances[29]. Detailed model specifications are provided in Supplementary Information and within our analysis scripts available at ref. 30.

Across all four conditions, groups began the game with an average cooperation rate of 69.5%. As expected, cooperation degrades substantially in the static network condition (generalized linear mixed model; coefficient −0.24, 95% confidence interval (CI) −0.27 to −0.20, $P < 0.001$; Fig. 2a). Without the opportunity to update their connections, groups quickly succumb to the tragedy of the commons: cooperation levels decline to 42.8% by the time the game ends in round 15. The random recommendation baseline (coefficient −0.13, 95% CI −0.16 to −0.10, $P < 0.001$; Fig. 2b) and cooperative-clustering baseline (coefficient −0.07, 95% CI −0.10 to −0.04, $P < 0.001$; Fig. 2c) mitigate the initial decline of cooperation, concluding the game with higher cooperation rates than observed on static networks. Nonetheless, cooperation still declines over time, ending at 57.0% with random recommendations and 61.2% with cooperative clustering.

In contrast, under the GraphNet social planner, cooperation rates increase significantly over the course of the game (coefficient 0.04, 95% CI 0.01 to 0.07, $P = 0.007$; Fig. 2d). Groups conclude the game in round 15 with a cooperation rate of 77.7%. Comparing directly against the other rewiring strategies, the GraphNet planner supports significantly higher rates of cooperation than static networks ($z = 13.0$, $P < 0.001$), random recommendations ($z = 8.3$, $P < 0.001$) and cooperative clustering ($z = 5.4$, $P < 0.001$), respectively (two-tailed comparisons, adjusted for multiple testing; Supplementary Fig. 6). To help illustrate the divergent outcomes fostered by the GraphNet and baseline planners, Fig. 2e provides graphical illustrations of networks from each condition. With the support of the GraphNet planner, groups enjoy high levels of capital relative to the other conditions (Extended Data Fig. 1), as well as minimal inequality (Fig. 2f; see also Extended Data Fig. 1).

To better understand the GraphNet planner's strategy, we analyse each planner's recommendations by valence (connect or disconnect) and by the cooperation decisions of the players involved (cooperate–cooperate, cooperate–defect or defect–defect). The random recommendation planner is not designed to take player choices into account when generating recommendations. Indeed, its behaviour in the actual groups provides no empirical evidence that participant choice affects its recommendations ($\chi^2(2) = 0.9$, $P = 0.639$; likelihood

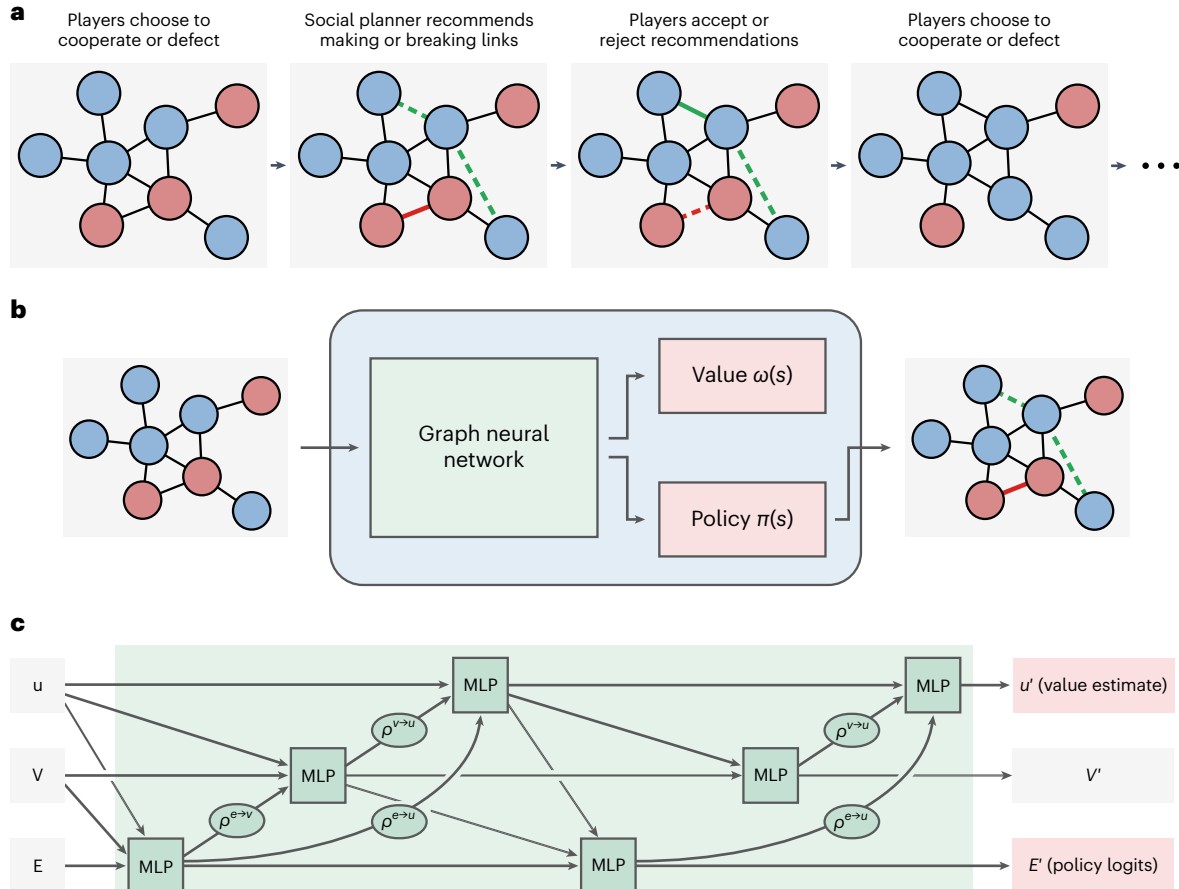

**Fig. 1 | Overview of the network cooperation game and our social-planning agent. a**, In this cooperation game, players are connected on a network and decide to cooperate with or defect from their neighbours. A social planner observes the players' decisions and the network structure, and then recommends changes to the network. Players choose to accept or reject the recommendations to their connections, and then a new turn begins. **b**, Our agent learns to act as the social planner and makes rewiring recommendations through a graph neural network (a 'GraphNet'). We optimize the GraphNet through reinforcement learning, producing a value function $\omega(s)$ and policy function $\pi(s)$. **c**, GraphNets explicitly encode graph structure in their computations. In this cooperation game, social planners observe the entire network of players and their cooperation decisions from the most recent round. The GraphNet in our social planning agent observes this information and processes the graph's global features ($u$), node features ($V$) and edge features ($E$) with a sequence of multilayer perceptrons (MLPs) and summation functions ($\rho$s), producing policy logits ($E'$) and a value estimate ($u'$). For more detail, see Supplementary Information Section E.

ratio test). The cooperative-clustering planner, in contrast, explicitly incorporates player choices into its planning algorithm: empirically, participant choices exert a significant influence on its recommendation patterns ($\chi^2(2) = 92.0$, $P < 0.001$).

We empirically find that the GraphNet planner learns a conditional approach to its recommendations, taking into account the cooperation decisions of the participants involved on each edge ($\chi^2(2) = 3451.8$, $P < 0.001$; Fig. 3). A representation analysis[31,32] provides convergent evidence that the social planner learns to encode and track the cooperativeness of the human participants in its neural network (Supplementary Fig. 7 and Supplementary Information Section F2).

The GraphNet planner virtually always recommends establishing links between cooperators ($P = 0.99$, 95% credible interval (CrI) 0.99 to 1.00), and rarely suggests removing them ($P = 0.03$, 95% CrI 0.03 to 0.03; Fig. 3a). The planner avoids creating new connections between defectors ($P = 0.00$, 95% CrI 0.00 to 0.00), and—unlike the cooperative-clustering baseline—recommends breaking existing links between defectors ($P = 1.00$, 95% CrI 0.99 to 1.00; Fig. 3c). This approach diminishes clustering among defectors. Defectors rarely connect with one another under the GraphNet planner, as exemplified in Fig. 2e. The GraphNet planner also suggests a mix of making connections

($P = 0.58$, 95% CrI 0.56 to 0.60) and breaking connections ($P = 0.50$, 95% CrI 0.49 to 0.52) involving one cooperator and one defector (Fig. 3b). The nuance of these cooperate–defect link recommendations becomes clearer when examining the planner's strategy over time. The GraphNet planner discovers a strategy that initially takes a conciliatory stance towards defectors, establishing a number of cooperate–defect links at the beginning of the game (Fig. 3d). As the game progresses, the GraphNet planner grows increasingly protective of cooperators, recommending a greater number of deletions for cooperate–defect links (Fig. 3e). The planner sends the average defector 1.4 recommendations (interdecile range 0–4) to connect with cooperators in each of the first four rounds, compared with 0.9 recommendations (interdecile range 0–3) in each of the last four rounds.

This conciliatory approach produces distinct patterns of network assortativity compared with the other conditions (Fig. 4a,b). In particular, the GraphNet planner induces near-zero assortment between cooperators and defectors (linear model; $\beta = -0.06$, 95% CI −0.14 to 0.02, $P = 0.142$; Fig. 4a). In contrast, and as expected, cooperative clustering induces positive choice assortativity by the end of the game, reflecting a positive tendency for cooperators to cluster with cooperators and defectors to cluster with defectors ($\beta = 0.10$, 95% CI 0.01 to 0.19,

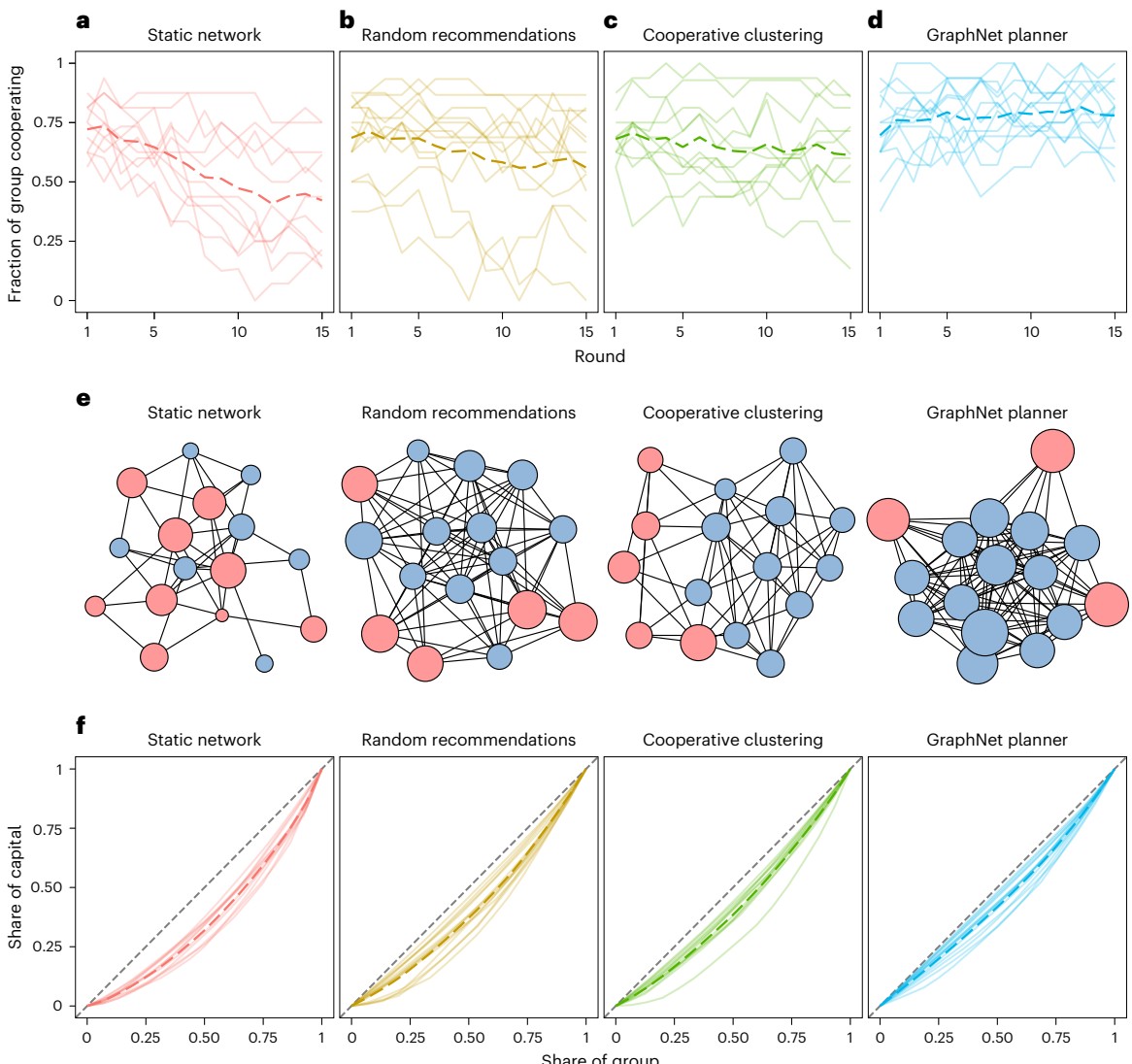

**Fig. 2 | Group outcomes fostered across different conditions.** Bold dotted lines represent the mean level across groups. Solid lines reflect the levels in individual groups. **a**, Cooperation levels tend to devolve in static networks. **b**, Random recommendations mitigate the decline of cooperation. **c**, Similarly, the cooperative-clustering social planner stabilizes cooperation levels. **d**, In contrast, the GraphNet social planner strengthens cooperation above starting levels. **e**, These networks illustrate representative games from round 10 of each condition. Node colour represents the participant's previous choice (blue, cooperate; red, defect). Node size reflects cumulative cooperative capital (larger nodes indicate a greater amount of capital). **f**, The GraphNet planner induces high levels of group equality. These Lorenz curves display the cumulative share of capital held by the group in the final round of the game, with the dashed 45° line reflecting perfect equality.

$P = 0.029$). These patterns are robust to multiple specifications for calculating assortative mixing (Supplementary Information Section F2 and Extended Data Fig. 2). Remarkably, the GraphNet planner's non-assortative strategy does not blunt the connectivity of cooperators (linear model; $\beta = 6.2$, 95% CI 5.3 to 7.2, $P < 0.001$; Fig. 4b). The relative degree for cooperators under the GraphNet planner significantly exceeds the levels seen on static networks ($t(44) = 9.0$, $P < 0.001$) under the random recommendations planner ($t(44) = 5.5$, $P < 0.001$) and under the cooperative-clustering planner ($t(44) = 5.4$, $P < 0.001$), respectively (two-tailed comparisons, adjusted for multiple testing). Under the GraphNet planner, participants enjoy non-assortative interactions, with cooperators exerting an outsize influence throughout the graph.

These patterns—non-assortativity and high connectivity for a subset of nodes in a graph—are characteristic of a core–periphery structure[33]. Consequently, we investigate the possibility that the GraphNet planner organizes communities into core–periphery networks. To

do so, we estimate the degree to which networks in each condition manifest a core–periphery structure (ref. 34; see also Supplementary Information Section F2). Groups receiving the GraphNet planner's recommendations exhibit significant levels of core–periphery structure (linear model; $\beta = 0.46$, 95% CI 0.35 to 0.58, $P < 0.001$). Within the GraphNet planner condition, cooperators account for on average 96.7% of the network core, and defectors 61.2% of the periphery. This pattern is extremely unlikely to emerge by chance ($P < 0.001$; permutation test). Rather than punish defectors with exclusion, the planner recommends they move into small, highly cooperative neighbourhoods (Fig. 4c). Visual inspection of networks formed by the different groups of participants underscores how consistently this core–periphery pattern emerges (Supplementary Fig. 8).

This approach represents a substantial departure from prior studies, in which 'decentralized ostracism' reduces the relative payoffs for defectors and—as the argument goes—incentivizes them to begin

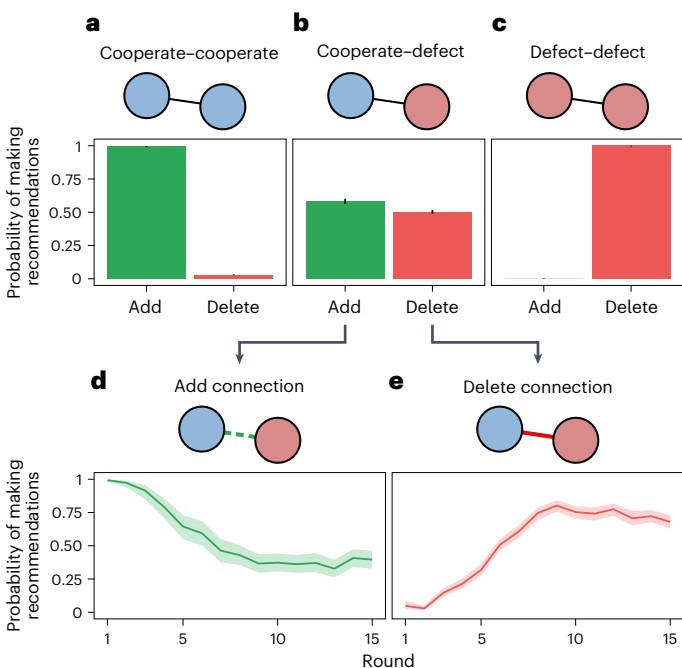

**Fig. 3 | Recommendations from the GraphNet planner.** The planner's recommendations varied as a function of the cooperation choices of the participants involved (cooperate–cooperate, cooperate–defect, or defect–defect) and the potential change to recommend (add or delete). Plots present mean values. Error bars and bands reflect 95% CrIs, with the posterior distribution computed from a uniform prior. Since the large number of observations precludes visualizing individual data points, we present the underlying data in Supplementary Table 6. **a**, The GraphNet planner virtually always recommends adding cooperator–cooperator links when it has the chance ($n = 1,854$ chances) and rarely suggests removing them ($n = 12,324$ chances). **b**, Uniquely, the GraphNet planner recommends a mixture of adding and removing cooperator–defect links ($n = 2,631$ chances and $n = 5,268$ chances, respectively). **c**, The planner also avoids recommending new defector–defector links ($n = 809$ chances). When given the chance, it virtually always suggests removing them ($n = 514$ chances). **d**, The GraphNet planner suggests adding a number of cooperator–defector links at the beginning of the game, but makes fewer of these recommendations over time ($n = 2,631$ chances over all rounds). **e**, Moving towards the end of the game, it increasingly recommends removing cooperator–defector connections ($n = 5,268$ chances over all rounds).

cooperating[14,15,28]. For example, cooperative clustering decreases the mean payoff for defectors over time, causing the relative payoff advantage of defection to gradually disappear. In contrast, under the Graph-Net social planner, the average payoff for defectors never declines below the average payoff for cooperation (Supplementary Fig. 9). In spite of this payoff gap, the planner is still able to maintain cohesive group cooperation, with minimal group inequality relative to the other conditions (measured through the Gini coefficient; Fig. 2f). Our application of deep reinforcement learning breaks from prior approaches and converges on an encouraging approach towards defectors.

Deep learning methods are often criticized for their lack of 'interpretability' (refs. 35,36; but see also ref. 37). In particular, it is difficult to know what exactly drives the behaviour of deep neural networks, given the inherent complexity, opacity, and high non-linearity of the solutions they learn (the 'black box' problem). We test whether the conciliatory patterns we observe are sufficient to explain the Graph-Net planner's performance. Alternatively, its success may stem from the black box of deep learning—that is, from a mechanism more difficult for us to interpret. To test these alternatives, we construct a new 'encouragement' social planner based on our analysis of the GraphNet planner's policy (that is, the patterns depicted in Fig. 3). In contrast

with the GraphNet planner's complex, opaque computations, the encouragement planner makes recommendations as a simple function of player cooperation choices and the round number (Supplementary Tables 7–9). For example, when faced with a connection between a cooperator and a defector in round 1, the encouragement planner will recommend removing the link with 4.8% probability; faced with a similar pair in the final round, it will recommend removing the link with 72.2% probability.

A follow-up study with human groups ($N = 224$ participants across 14 groups) validates the effectiveness of the conciliatory approach we observe from the GraphNet planner. The encouragement planner significantly improves group cooperation levels over the course of the game (generalized linear mixed model; coefficient 0.04, 95% CI 0.00 to 0.06, $P = 0.005$; Fig. 5a). A direct comparison shows that the encouragement approach significantly outperforms static networks ($z = 13.4$, $P < 0.001$), random recommendations ($z = 8.4$, $P < 0.001$) and cooperative clustering ($z = 5.4$, $P < 0.001$) at supporting group cooperation, respectively (two-tailed comparisons, adjusted for multiple testing; Supplementary Fig. 10). The encouragement planner enhances group cooperation to a similar extent as the GraphNet planner ($z = -0.3$, $P = 1.000$) and exerts similar effects on network assortativity (Fig. 5b), consistently engineering a core–periphery structure for groups (Fig. 5c; see also Supplementary Fig. 12).

The recommendations from both the GraphNet planner and the encouragement planner produce networks with notably high density, especially relative to the baseline conditions (Extended Data Fig. 1). Under the GraphNet planner, for example, several groups reached full networked connectivity (Supplementary Fig. 8). To evaluate the possibility that high density alone drives the success of these planners—without the need for an encouraging approach—we run two additional follow-up studies ($N = 400$ participants across 25 groups). First, we build a 'neutral' social planner that aims to recreate the connectivity dynamics observed under the GraphNet planner, without regard for player's choices (that is, dispensing with the encouraging approach to defectors; $N = 192$ participants across 12 groups). As intended, this planner generates levels of network connectivity to a similar extent as the GraphNet planner ($t(80) = -1.93$, $P = 0.468$; two-tailed comparison). Nonetheless, its choice-agnostic approach degrades group cooperation significantly over time (generalized linear mixed model; coefficient $-0.17$, 95% CI $-0.19$ to $-0.14$, $P < 0.001$). As a further test of whether network density drives the high cooperation rates seen with the GraphNet planner, we construct another social planner that seeks to maximize network connectivity as much as possible ($N = 208$ participants across 13 groups). This strategy generates levels of network density that significantly exceed those produced by the GraphNet planner ($t(80) = 5.34$, $P < 0.001$; two-tailed comparison). However, this also causes a precipitous decline in cooperation (coefficient $-0.51$, 95% CI $-0.55$ to $-0.46$, $P < 0.001$). On its own, network density does not offer a compelling explanation for the high cooperation rates supported by the GraphNet planner.

Overall, these three follow-up studies help to validate the value and sufficiency of an encouraging approach to defectors.

## Discussion

How can a social planner best support group cooperation and mitigate the spread of defection? Prior methods focus on increasing the assortment of strategy types within a networked group. This approach protects cooperators from antisocial contagion and simultaneously punishes defectors for their selfishness.

We build a social planner that learns for itself how to scaffold cooperation, through deep reinforcement learning and repeated trial and error in simulation. Our social planner proves capable of not only stabilizing, but also enhancing cooperation over time. The planner's strategy validates several characteristics of prior approaches, including a tendency for cooperators to connect with other cooperators. It does not, however, partition defectors away from cooperators ('decentralized

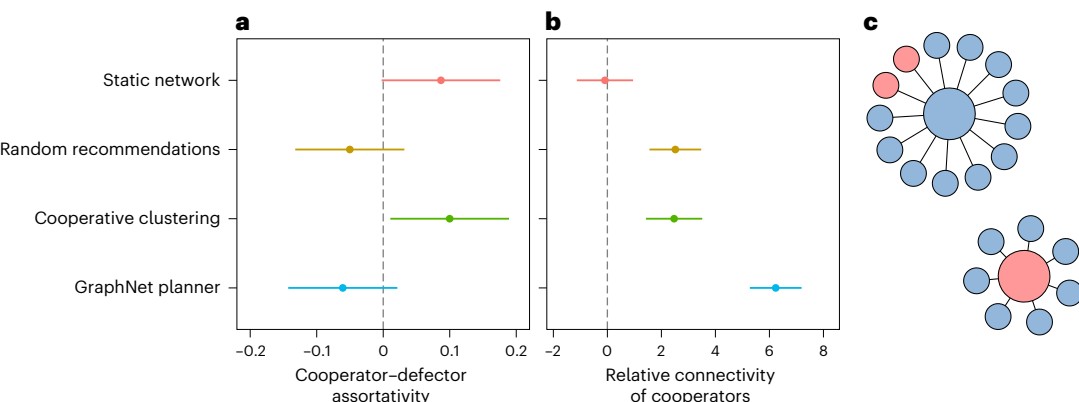

**Fig. 4 | Mixing patterns induced by the different social planners.** Plots present effect estimates from linear models. Error bars indicate 95% CIs. **a**, Network rigidity and cooperative clustering produce significant choice assortativity (visualized here at the end of the game, with $n = 48$ groups over all conditions). Cooperators tend to connect with cooperators, and defectors with defectors. The GraphNet planner, in contrast, induces non-assortativity between cooperators and defectors. **b**, The GraphNet planner maximizes the relative connectivity of cooperators in the network game, as measured by the difference in the average degree of cooperating participants and defecting participants

(that is, the mean degree bias towards cooperators; visualized for the final round, with $n = 48$ groups over all conditions). **c**, The mixing patterns engineered by the GraphNet precipitate drastically different experiences for cooperators and defectors. On expectation, cooperators inhabit large neighbourhoods with a mix of cooperators and defectors. In contrast, defectors experience small neighbourhoods with virtually only cooperators. The neighbourhoods here depict the median cooperator and defector counts for cooperators' and defectors' neighbourhoods partway through the game, on round 10.

ostracism'[14,15,28]). Instead, the planner recommends a core–periphery structure for the community. Though defecting participants move to the periphery of the graph, they remain well connected to cooperators. This encouraging, conciliatory approach fosters pro-social contagion while minimizing the spread of defection. Echoing dynamics observed in collective action[38], a critical mass of cooperative individuals can draw cynical outsiders into the fold.

Prior studies in this domain emphasize higher relative payoffs for cooperation as central to incentivizing players to abandon defection[28,39]. Our social planner succeeds at encouraging group cooperation, but unexpectedly, the networks that it engineers consistently reward defectors more than cooperators. This discrepancy indicates that short-term utility calculus can provide only a partial explanation for participants' behaviour in this network game. Future studies should draw inspiration from psychology research to better understand participants' motivation and thinking. Non-economic factors such as preferences for fairness[40] and conformity[41,42] probably contribute to the contagiousness of cooperation. Overall, deep reinforcement learning discovers a novel approach to the challenge of scaffolding community cooperation.

Our social planner learns to make its recommendations through a graph neural network. Multiple experiments demonstrate the effectiveness of graph neural networks in solving physical problems (for example, refs. [43–45]). Notably, social scientists have long argued that social systems are well modelled through physics ('social physics'[46]). This consonance may explain the GraphNet planner's effectiveness, and additionally suggests that our approach may prove applicable to other graphical games[47] modelling community dilemmas. The combination of graph neural networks, reinforcement learning, and simulation could uncover novel solutions to challenges such as resource sharing[48] and efficient innovation and discovery[49,50].

Developments in machine learning indicate several promising directions for future research. The design of graph neural networks allows them to generalize to large-scale problems. Several teams, for example, have applied graph neural networks to so-called 'web-scale' challenges, involving millions of nodes and potentially billions of edges[51,52]. These successes hint at a path to scaffolding cooperation in expansive networks: can an encouraging approach support community cohesion at large scales? Another potential path concerns interpretability. Recent work demonstrates that large language models (for

example, refs. [53,54]) may be capable of generating explanations for algorithmic decision making[55]. With the support of a language model, our social planner could explain its policy to group members in natural language, helping them to understand the possible consequences of any choices that they might make.

Ethicists and policymakers emphasize human autonomy as a central value for the development and deployment of AI[56,57]. Nonetheless, modern AI research does not always afford human participants much control or power within the context of their interaction with AI systems. In our experiments, our agent's actions are entirely recommendation based: participants have the option to accept or reject the decisions that the agent makes. These decisions to accept or reject system advice reflect a revealed preference within human–AI interaction[58]. In addition to recommendation-based approaches, future interaction research can support autonomy through other revealed-preference frameworks, perhaps including the choice of entirely opting out of interactions with the agent in question (an 'exit option'[59]). The deployment of agents to assist with social planning raises additional questions concerning consent and governance. Which stakeholders should direct, steer and fund AI systems in this domain? The application of participatory and democratic methods will be particularly important for such technology[60,61]. It is imperative that technologists preserve the ability of communities that will be affected by AI to engage with it on their own terms—whether that is to withdraw from, contribute to, steer or potentially resist the deployment of these systems.

AI increasingly infuses everyday life. As a result, people enjoy a growing range of relationships with AI systems, forming 'hybrid societies' of human and algorithmic actors[62,63]. Some applications of AI technology call for a physical, embodied presence to interact with humans[15,64,65]. Others, like the algorithmic social planner in our study, may be less visible to the communities with which they interact, yet no less influential. Both categories merit expanded research and study. Overall, our results contribute to a growing body of evidence that agents trained with deep reinforcement learning can enhance collaboration and cooperation[20,21,66–68]. AI can prove a positive, beneficial force to support human communities.

## Methods

Our research complies with all relevant ethical regulations. The experimental protocol underwent independent ethical review and received

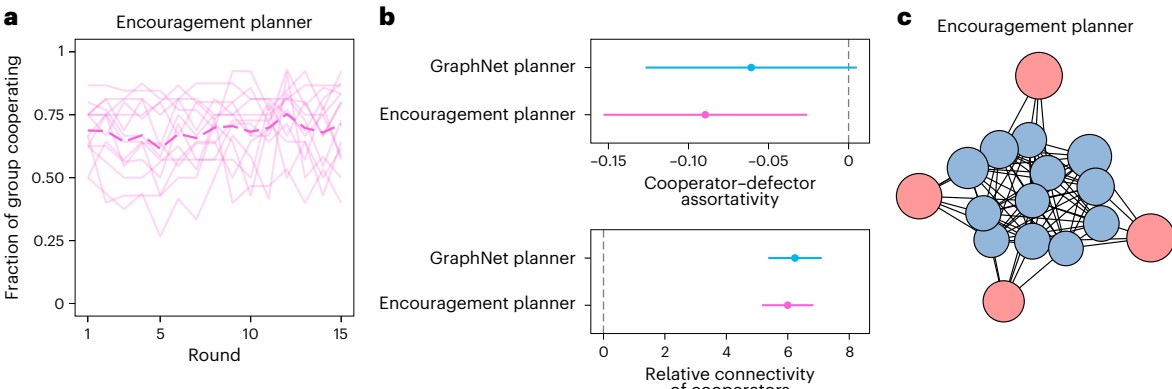

**Fig. 5 | Group cooperation levels and network structure cultivated by the 'encouragement' social planner.** Based on our analysis of the GraphNet planner's conciliatory approach to defectors, the encouragement planner makes recommendations as a simple function of player cooperation choices and the round number. **a**, The encouragement planner stabilizes cooperation levels among human groups. The bold dotted line indicates the mean level across groups. Solid lines depict the levels in individual groups. **b**, The encouragement planner reproduces the patterns of assortativity engineered by the GraphNet planner (visualized here for the final round, with $n = 27$ groups over both conditions). Plots present effect estimates from linear models. Error bars indicate 95% CIs. **c**, This network shows a representative game under the encouragement planner in round 10. Like the GraphNet planner, the encouragement planner tends to recommend a core–periphery structure for groups.

a favourable opinion from the Human Behavioural Research Ethics Committee at Google DeepMind (#19/004).

We trained an artificial agent to act as a social planner in the cooperative network game through reinforcement learning and simulation methods. The agent comprised a graph neural network (a 'GraphNet'[26]) with two message-passing steps (Fig. 1c). The architecture was non-recurrent. The agent optimized for a combination of capital level and recommendation quality (Supplementary Information Section E3), and used advantage actor–critic[27] as its learning algorithm over a distributed framework[69]. For more detail on the agent design and parameterization, see Supplementary Information Section E.

We constructed bots to simulate human cooperation and recommendation acceptance decisions for the agent's training. Each bot $i$ randomly sampled a cooperative disposition parameter, $\theta_i \sim \mathcal{N}(\mu_\theta, \sigma_\theta)$, upon its initialization. Bots made cooperation choices through two logistic functions, conditional on the current round number $t$. In the initial round, when a bot had no information about the behaviour of its neighbours, it randomly sampled an action (cooperate or defect) as a logistic function of its disposition parameter $\theta_i$ and two parameters shared by all bots: $\beta_0'$ and $\beta_1'$. In subsequent rounds, the bot chose to cooperate as a logistic function of its current neighbourhood size $x_s$, its current number of cooperating neighbours $x_n$, the current rate of cooperation in its neighbourhood $x_r$, its disposition parameter $\theta_i$, and four parameters shared by all bots: $\beta_0, \beta_1, \beta_2$ and $\beta_3$:

$$P_{\text{cooperate}}(t, i) = \begin{cases} \dfrac{1}{1+e^{-(\beta_0' + \beta_1' \cdot \theta_i)}} & \text{if } t = 1 \\[2ex] \dfrac{1}{1+e^{-(\beta_0 + \beta_1 \cdot x_s + \beta_2 \cdot x_n + \beta_3 \cdot x_r + \theta_i)}} & \text{otherwise} \end{cases}$$

The bot accepted or rejected recommendations from the social planner as a function of the recommendation valence $a_{\text{SP}}(i,j) \in \{-1 \text{ to } 1\}$ (where −1 signifies 'break link' and 1 reflects 'make link') and the referent neighbour's previous cooperation decision $a_j^0 \in \{0,1\}$ (where 0 denotes defection and 1 represents cooperation):

$$P_{\text{accept}}\left(a_{\text{SP}}(i,j), a_j^0\right) = \begin{cases} \varphi_0 & \text{if } a_{\text{SP}}(i,j) = -1 \text{ and if } a_j^0 = 0 \\ \varphi_1 & \text{if } a_{\text{SP}}(i,j) = -1 \text{ and if } a_j^0 = 1 \\ \varphi_2 & \text{if } a_{\text{SP}}(i,j) = 1 \text{ and if } a_j^0 = 0 \\ \varphi_3 & \text{if } a_{\text{SP}}(i,j) = 1 \text{ and if } a_j^0 = 1 \end{cases}$$

To select the $\mu_\theta, \sigma_\theta, \beta$ and $\varphi$ parameters for the bots, we fit models to behavioural data collected in the baseline conditions of the group experiments. For fitted values and more information on the bot design, see Supplementary Information Section E5.

We trained 30 replicates of the agent for a maximum of $5 \times 10^7$ simulated game rounds, using a different random initialization for the neural network in each replicate. Across multiple random network initializations, the social planner learned qualitatively and quantitatively similar policies that scaffolded high levels of cooperation with simulated groups. We selected one of these high-performing policies to evaluate with human participants.

We recruited participants from Prolific[70] for our group experiments. All participants provided informed consent before joining the study. In addition to the summary provided here, see Supplementary Information Section C for full details of the study design. The experiments employed a between-participants design: that is, participants joined a single group (with no participant experiencing multiple conditions). The experiments were also incentive compatible: that is, participants (knowingly) made decisions in the cooperative network game for real monetary stakes. Participants first read detailed study and game instructions, played a short tutorial round, and subsequently completed a comprehension test on the game rules. We required participants to answer all three questions correctly to continue. The majority (74.2%) passed the test and were randomly sorted into groups of $n = 16$ participants each. We provided the remainder a show-up payment for their time. The final sample comprised $N = 1,392$ participants (mean age of 36.7 years, standard deviation 12.7 years; 44.9% female, 52.6% male and 1.4% non-binary, trans, genderqueer, demigender, agender, asexual and aromantic). For the demographics of the baseline conditions ($N = 560$), the evaluation condition ($N = 208$) and the validation conditions ($N = 624$), see Supplementary Information Sections D2, F1 and G2, respectively.

Each group consisted of 16 participants and played 15 rounds of the cooperative network game (Supplementary Figs. 29–35). To avoid end-game effects, participants were not told how many rounds to expect. Each stage of the game (for example, choosing to cooperate or receiving recommendations from the planner) waited a pre-set amount of time for participant input. Participants that did not respond were removed from the experiment. We subsequently provided these participants with a debrief questionnaire including questions about any technical problems they may have encountered. The tutorial explicitly

detailed these rules for participants, and the main game interface displayed a timer at the bottom of every page counting down the time remaining for the current choice. Empirically, the groups experienced a very low dropout rate among participants, ending with a mean of 14.6 participants (median 15). After completing the game, participants completed a short questionnaire and then received their compensation for the study. Participants completed the study in an average of 26.5 min and earned an average overall payment of US$11.79 for participating.

We processed data from the group experiments using Python 3.9.15 and conducted data analysis using R 4.1.3.

### Reporting summary

Further information on research design is available in the Nature Portfolio Reporting Summary linked to this article.

## Data availability

The data necessary for reproducing all analyses and figures are available at https://osf.io/8ahkg/.

## Code availability

The analysis scripts necessary for reproducing all analyses and figures are available at https://osf.io/8ahkg/.

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

## Acknowledgements

We thank Z. Ahmed, F. Fischer, M. Patel and J. Sanchez Elias for providing technical support for the project; S. Mohamed for offering general support; and I. Gemp, G. Reinecke, B. Tracey and K. Tuyls for offering feedback on the manuscript. The authors received no specific funding for this work.

## Author contributions

K.R.M. proposed the research idea; K.R.M. and M.B. designed the research; K.R.M. coded and developed the baselines and agents, with assistance from A.T., J.B. and R.E.; M.A.B. independently coded and verified the baselines and agents; K.R.M. coded the study, collected data and conducted statistical analysis; K.R.M. interpreted results, with assistance from A.T., M.A.B., J.B., L.C.-G., R.E. and M.B.; and K.R.M. wrote the paper, with assistance from A.T., M.A.B., J.B., L.C.-G., R.E. and M.B.

## Competing interests

The authors declare no competing interests.

## Additional information

**Extended data** is available for this paper at https://doi.org/10.1038/s41562-023-01686-7.

**Correspondence and requests for materials** should be addressed to Kevin R. McKee.

## a  Average cooperation rate in group

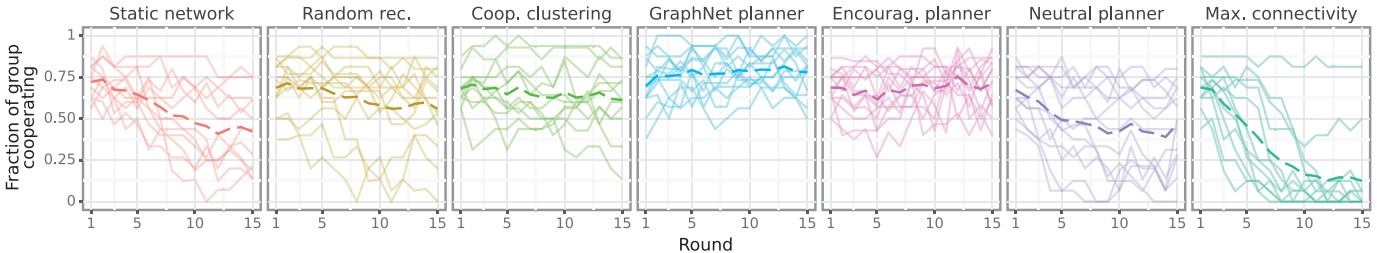

## b  Group connectivity

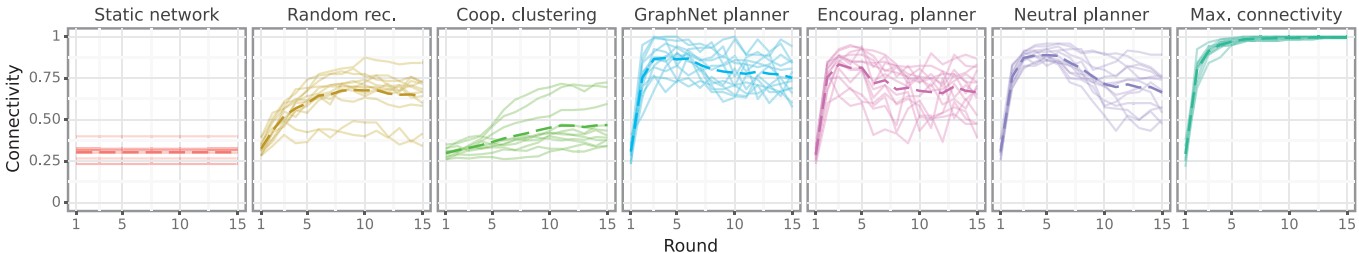

## c  Average accumulated capital in group

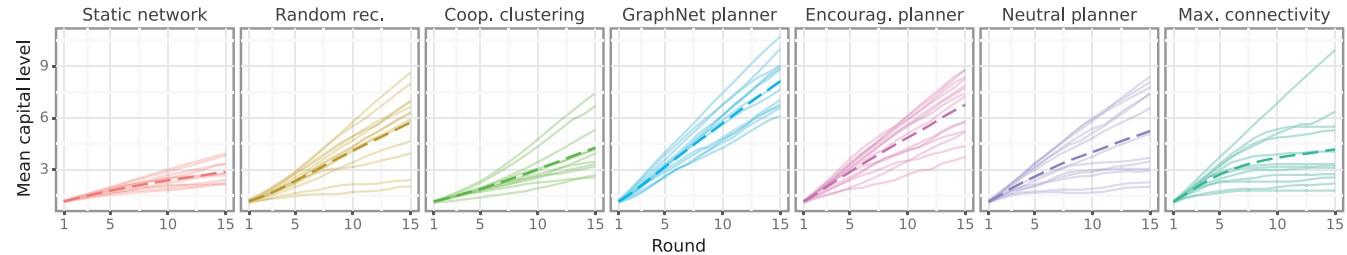

## d  Inequality of capital distribution within group

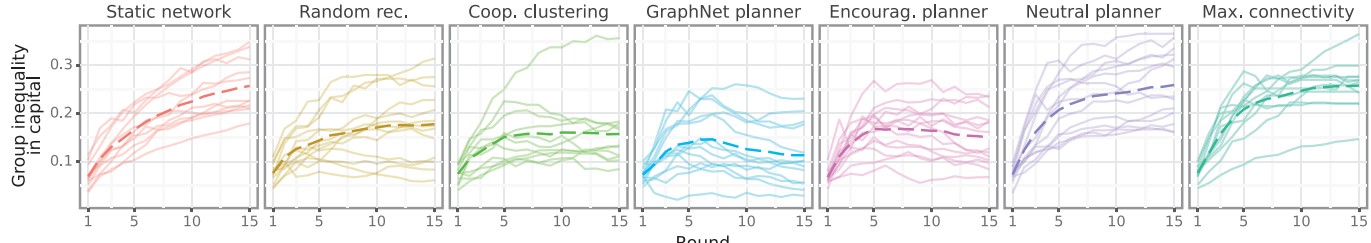

**Extended Data Fig. 1 | Group outcomes over time.** Bold dotted lines represent the mean level across sessions. Solid lines reflect the levels in individual sessions. **a**, Group cooperation rate is calculated as the fraction of the group choosing to cooperate. **b**, Group connectivity is calculated as the fraction of group members linked to one another, relative to the total number of possible connections. **c**, Group capital level is calculated as the average accumulated capital level across all group members. **d**, Group inequality is calculated as the Gini coefficient of accumulated capital levels across all group members.

**a** Assortative mixing between cooperators and defectors

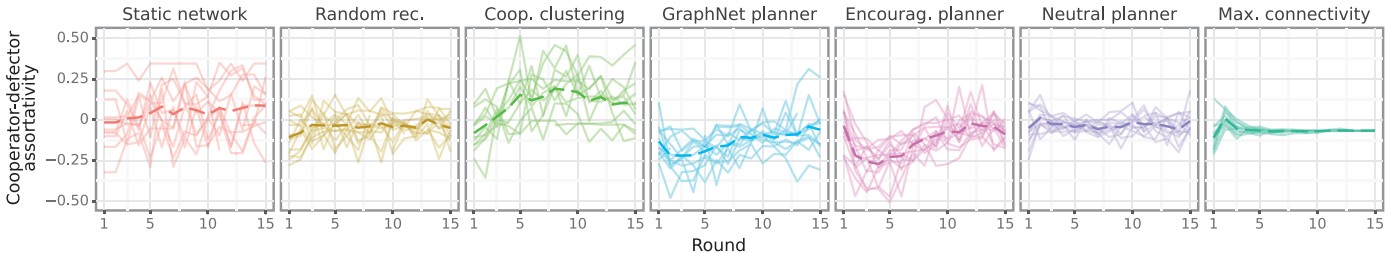

**b** Disassortative separation between defectors

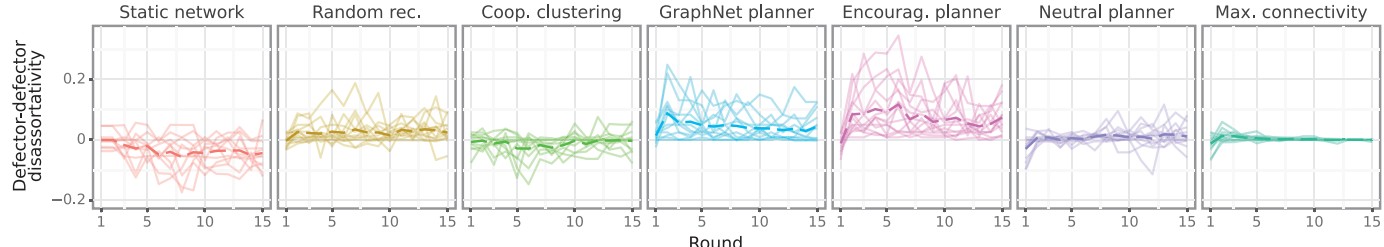

**c** Relative degree of cooperators

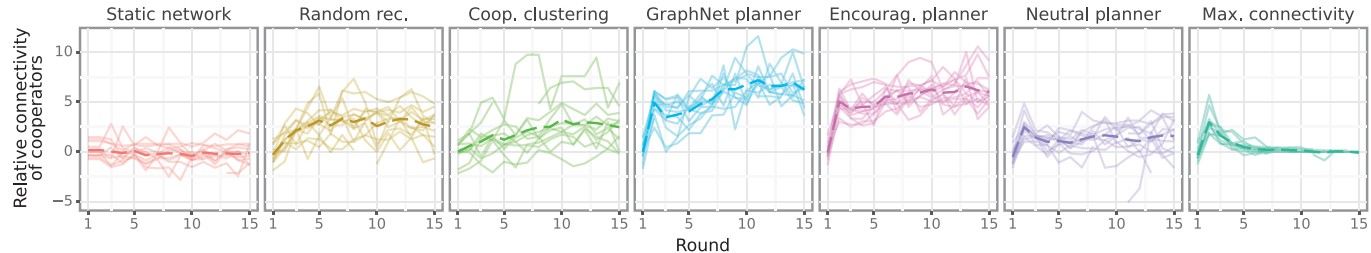

**d** Core-periphery structure

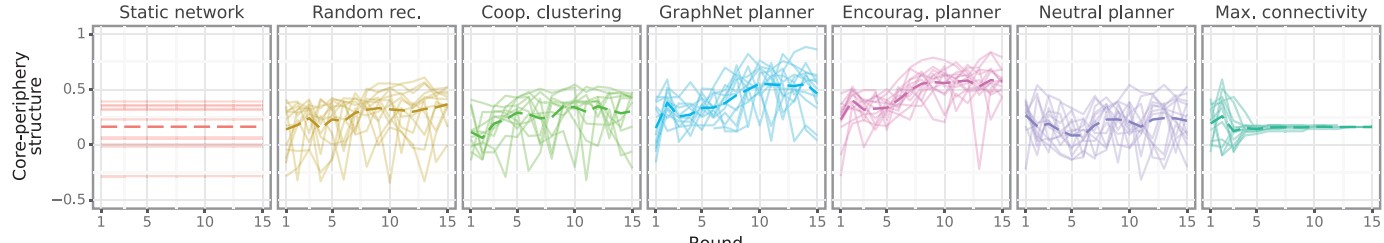

**Extended Data Fig. 2 | Network outcomes over time.** Bold dotted lines represent the mean level across sessions. Solid lines reflect the levels in individual sessions. **a**, Assortative mixing between cooperators and defectors is calculated as the assortativity coefficient for cooperation choices across the network. **b**, Disassortative separation between defectors is calculated as the difference between the fraction of defector–defector links in the network expected under a random permutation of cooperation choices (estimated over 10,000 random replicates) and the actual observed fraction. **c**, Relative degree of cooperators is calculated as the mean degree of cooperators less the mean degree of defectors across the network. **d**, Core-periphery structure is calculated as the correlation between the adjacency matrix of the actual network structure and the idealized core-periphery adjacency matrix (based on the best-fitting classification of group members into core and periphery).

# Reporting Summary

## Statistics

For all statistical analyses, confirm that the following items are present in the figure legend, table legend, main text, or Methods section.

| n/a | Confirmed | |
|---|---|---|
| ☐ | ☒ | The exact sample size (*n*) for each experimental group/condition, given as a discrete number and unit of measurement |
| ☐ | ☒ | A statement on whether measurements were taken from distinct samples or whether the same sample was measured repeatedly |
| ☐ | ☒ | The statistical test(s) used AND whether they are one- or two-sided <br> *Only common tests should be described solely by name; describe more complex techniques in the Methods section.* |
| ☐ | ☒ | A description of all covariates tested |
| ☐ | ☒ | A description of any assumptions or corrections, such as tests of normality and adjustment for multiple comparisons |
| ☐ | ☒ | A full description of the statistical parameters including central tendency (e.g. means) or other basic estimates (e.g. regression coefficient) AND variation (e.g. standard deviation) or associated estimates of uncertainty (e.g. confidence intervals) |
| ☐ | ☒ | For null hypothesis testing, the test statistic (e.g. *F*, *t*, *r*) with confidence intervals, effect sizes, degrees of freedom and *P* value noted <br> *Give P values as exact values whenever suitable.* |
| ☐ | ☒ | For Bayesian analysis, information on the choice of priors and Markov chain Monte Carlo settings |
| ☒ | ☐ | For hierarchical and complex designs, identification of the appropriate level for tests and full reporting of outcomes |
| ☒ | ☐ | Estimates of effect sizes (e.g. Cohen's *d*, Pearson's *r*), indicating how they were calculated |

*Our web collection on statistics for biologists contains articles on many of the points above.*

## Software and code

Policy information about availability of computer code

| Data collection | We collected data using custom code. The code runs a platform combining standard questionnaire functionality with the ability to run games for both human participants and AI systems. |
|---|---|
| Data analysis | We processed data using Python 3.9.15 and analyzed data using R 4.1.3. Analysis scripts are publicly available via an OSF repository at https://osf.io/8ahkg/. |

For manuscripts utilizing custom algorithms or software that are central to the research but not yet described in published literature, software must be made available to editors and reviewers. We strongly encourage code deposition in a community repository (e.g. GitHub). See the Nature Portfolio guidelines for submitting code & software for further information.

## Data

Policy information about availability of data

All manuscripts must include a data availability statement. This statement should provide the following information, where applicable:
- Accession codes, unique identifiers, or web links for publicly available datasets
- A description of any restrictions on data availability
- For clinical datasets or third party data, please ensure that the statement adheres to our policy

Data are publicly available via an OSF repository at https://osf.io/8ahkg/.

# Human research participants

Policy information about [studies involving human research participants and Sex and Gender in Research.](studies involving human research participants and Sex and Gender in Research.)

| | |
|---|---|
| Reporting on sex and gender | We asked participants to report their gender identity if they felt comfortable providing this information, for the purpose of understanding the representativeness of our sample. Our sample included comparable proportions of men and women, as well as a number of participants with other gender identities. See information in "Research sample" below. |
| Population characteristics | See information in "Research sample" below. |
| Recruitment | We recruited participants online, through the Prolific platform (https://prolific.co/). We published a study with the following inclusion criteria: residence in the U.S.; completion of at least 20 previous studies; approval rate of 95% or more on previous studies. Any Prolific participant who met those criteria could join our study.<br><br>One possible self-selection bias is toward extroverted or social individuals. We provided potential participants with an upfront description of the group nature of our study: "In this study, you will make decisions while interacting with other participants." |
| Ethics oversight | The Human Behavioural Research Ethics Committee (HuBREC) at Google DeepMind conducted independent review and oversight for our research. HuBREC is an ethics review board that provides independent review and oversight for human-participant research, staffed and chaired by academics from outside of Google DeepMind. |

Note that full information on the approval of the study protocol must also be provided in the manuscript.

# Field-specific reporting

Please select the one below that is the best fit for your research. If you are not sure, read the appropriate sections before making your selection.

☐ Life sciences    ☒ Behavioural & social sciences    ☐ Ecological, evolutionary & environmental sciences

For a reference copy of the document with all sections, see [nature.com/documents/nr-reporting-summary-flat.pdf](nature.com/documents/nr-reporting-summary-flat.pdf)

# Behavioural & social sciences study design

All studies must disclose on these points even when the disclosure is negative.

| | |
|---|---|
| Study description | The study used an experimental, between-participant design and collected quantitative outcome data. After participants completed an instructional tutorial and passed a comprehension test, we randomly assigned them into groups of 16 to play a cooperative network game. Each group played the game with one of seven "social planners." We measured various individual and group outcomes during the game, including cooperation choices, to compare the effectiveness of different social planners at encouraging cooperation. |
| Research sample | We recruited participants from the online platform Prolific, with the inclusion criteria of residence in the U.S. and completion of at least 20 previous studies with an approval rate of 95% or more. We collected demographic data on age, gender identity, and education. Based on the prior papers that established the experimental protocol for this study (Rand et al., 2011; Shirado et al., 2013), we aimed to recruit around 200 participants per condition. All participants provided informed consent before joining the study. The final sample comprised N = 1392, participants (mean age of 36.7, sd = 12.7; 44.9% female, 52.6% male, and 1.4% non-binary, trans, genderqueer, demigender, agender, asexual, and aromantic). |
| Sampling strategy | The study used convenience sampling from Prolific, an online recruitment platform. We determined sample size per condition based on the prior papers that established the experimental protocol for this study (Rand et al., 2011; Shirado et al., 2013), aiming to recruit around 200 participants per condition. |
| Data collection | Participants completed the study online, through a browser-based interface. The researchers were not blind to the study conditions, but—aside from providing troubleshooting information to participants with technical issues—did not interact with participants while they completed the study. |
| Timing | We collected data from May 19-24, 2021, Apr 7-25, 2022, and Mar 27, 2023. |
| Data exclusions | No participants were excluded from analysis. |
| Non-participation | On average, 1.4 participants dropped out of each study session of 16 participants (median = 1 drop-out per session). Participants dropped out automatically after failing to respond for a set amount of time during the study. |
| Randomization | Participants were randomly allocated to study sessions. |

# Reporting for specific materials, systems and methods

We require information from authors about some types of materials, experimental systems and methods used in many studies. Here, indicate whether each material, system or method listed is relevant to your study. If you are not sure if a list item applies to your research, read the appropriate section before selecting a response.

| Materials & experimental systems | Methods |
|---|---|

| n/a | Involved in the study |
|---|---|
| ☒ | Antibodies |
| ☒ | Eukaryotic cell lines |
| ☒ | Palaeontology and archaeology |
| ☒ | Animals and other organisms |
| ☒ | Clinical data |
| ☒ | Dual use research of concern |

| n/a | Involved in the study |
|---|---|
| ☒ | ChIP-seq |
| ☒ | Flow cytometry |
| ☒ | MRI-based neuroimaging |

