## [Peer Review File · Nature Human Behaviour]

Peer Review Information

Journal: Nature Human Behaviour

Manuscript Title: Scaffolding cooperation in human groups with deep reinforcement learning

Corresponding author name(s): Kevin R. McKee

Reviewer Comments & Decisions:

Decision Letter, initial version:

3rd February 2023

Dear Mr McKee,

Thank you once again for your manuscript, entitled "Scaffolding cooperation in human groups with deep reinforcement learning," and for your patience during the peer review process.

Your manuscript has now been evaluated by 3 reviewers, whose comments are included at the end of this letter. Although the reviewers find your work to be of interest, they also raise some important concerns. We are very interested in the possibility of publishing your study in Nature Human Behaviour, but would like to consider your response to these concerns in the form of a revised manuscript before we make a decision on publication.

In our view, the two most important issues to address will be Reviewer #3's point 1 (over whether the results are trivially due to increased network density) and point 8 on the statistical analysis.

In sum, we invite you to revise your manuscript taking into account all reviewer and editor comments. We are committed to providing a fair and constructive peer-review process. Do not hesitate to contact us if there are specific requests from the reviewers that you believe are technically impossible or unlikely to yield a meaningful outcome.

We hope to receive your revised manuscript within two months. I would be grateful if you could contact us as soon as possible if you foresee difficulties with meeting this target resubmission date.

- Include a "Response to the editors and reviewers" document detailing, point-by-point, how you addressed each editor and referee comment. If no action was taken to address a point, you must provide a compelling argument. When formatting this document, please respond to each reviewer comment individually, including the full text of the reviewer comment verbatim followed by your response to the individual point. This response will be used by the editors to evaluate your revision and sent back to the reviewers along with the revised manuscript.
- Highlight all changes made to your manuscript or provide us with a version that tracks changes.

[REDACTED]

We look forward to seeing the revised manuscript and thank you for the opportunity to review your work. Please do not hesitate to contact me if you have any questions or would like to discuss these revisions further.

Sincerely,
Jamie

Dr Jamie Horder
Senior Editor
Nature Human Behaviour

Reviewer expertise:

Reviewer #1: networks, cooperation

Reviewer #2: machine learning

Reviewer #3: networks, cooperation

REVIEWER COMMENTS:

Reviewer #1:
Remarks to the Author:

This is a nifty paper on a profound topic, namely: how cooperation may be sustained within human

groups, so as to avoid a tragedy of the commons that is so widely observed. It involves yet another challenging and important application of the sorts of artificial intelligence tools being developed at Deep Mind. The paper is just an initial foray into this topic (one that is, at least in this instantiation, admittedly less challenging, obviously, than AlphaGo or protein folding); but the paper is likely to push further an emerging body of work on the uses of AI to modify human group behavior. As such, it is likely to be of great interest to the readers of Nature Human Behavior. And the paper is well written and well executed.

My mostly small comments below are directed at clarifying some of the analysis and arguments.

I think it would be a bit clearer if the abstract (and rest of the paper) used the word "groups" instead of "sessions" to describe the various experiments.

At page 18, it is noted that the "social planner" (the AI agent) sits outside of and can observe the entirety of the social network as well as the cooperation decisions and capital levels of all the players. I think the body of the manuscript (for instance at page 2) should make it a bit clearer that the AI agent here has such global knowledge about the whole network of people and all their actions. To be clear, I like the contrast noted explicitly in footnote 2 about the difference between a distributed embodied approach and a centralized non-embodied approach. But this reality prompts the following concern: how this could scale with network group size? Is there any way to estimate how scalable this is? One of the reasons, I gather, that the Shirado iScience paper uses multiple decentralized bots is precisely to avoid the demanding requirement of such global surveillance.

Another subtlety that I think needs to be noted (for instance at page 3) is how many times steps backwards the planner observes the "players' recent decisions." If it is the entire history of the players, this is also another demanding feature. At page 5, for example, it is made quite explicit that the planner changes its behavior as the game progresses – which highlights the temporality concerns again. Just like the issue of how the challenge would scale with increasing group size, one has to wonder how the AI planner would scale with increasing time steps and duration of interactions?

I don't necessarily have a problem with the planner being trained on simulated human groups, especially given the sophistication of this Deep Mind team, but I couldn't help but wonder why the planner was not trained on real human groups?

For what it's worth, there are some important technical details regarding the implementation of the "cooperative clustering" algorithm (at page 20) that are different between this paper and the Shirado iScience paper. But these differences really don't matter for the argument this paper is making.

I really like the observation about how the planner achieves core-periphery structure rather than simply clustering the defectors together on the edge of the graph (as noted on pages 7 and 8).

As I understand the modeling approach, the P values reported on page 6 and elsewhere arise from comparing individuals, not groups. That is, the sample size is the total number of people, not the total number of groups. This should be clarified. Also, does this statistical model described at the bottom of page 20 include group-level fixed effects?

The statistical model described to page 9 has both the current rate of cooperation and the number of

cooperating neighbors in the same model. I can understand why this parameterization was chosen, but surely those two variables are extremely collinear? At least that should be discussed?

The network group structure parameters in table S1 are standard and appropriate.

A bit more information on how nodes (subjects) that drop out are handled would be appreciated in the SI. I suspect that the investigators do not replace the dropouts with bots nor do they rewire the network around the dropped players. Probably they have the dropped player simply continue to play their last play (i.e., cooperation or defection)? But then what do the players do when offered the opportunity to make or cut ties by the planner? Maybe this was mentioned somewhere and I missed it?

Another paper the authors might possibly wish to take a look at is this 2012 paper by Apicella et al. on cooperative clusters in Hadza hunter gathers: <https://www.nature.com/articles/nature10736>

Reviewer #2:
Remarks to the Author:

Summary of findings.

The paper addresses an important open question: how to encourage group cooperation and mitigate the spread of defection across the population. In their model, the authors present a social planner, which makes recommendations to rewire their ties with others in the network based on player's past actions as well as the network activity.

Fit to Nature Human Behaviour:

This research contribution is similar to the ones made by Shirado and Christakis, which got published in venues that are comparable Nature Human Behaviour (NHB). Moreover, the paper is generally well written, easy to follow, and addresses topics of broad appeal, making the paper suitable to a general audience. Hence, I believe the papers fits the scope of NHB.

Novelty:

While the paper does not make a theoretical contribution towards our understanding of human behavior, nor identifies a novel mechanism to establish cooperation, it does make a significant contribution towards understanding the potential use of AI to help solve social dilemmas. The authors neatly combine existing research to propose a solution, which uses Artificial Intelligence (AI) to facilitate cooperation, building on a similar line of research by Rand et al. [2011] as well as Shirado and Christakis [2020]. Importantly, unlike previous solutions, the authors' social planner is able to increase cooperation levels (rather than reduce the decline in cooperation levels) over the course of the game. I liked how the social planner learns a strategy from scratch, through repeated trial and error. I also like how, instead of ostracizing defectors (as is the case with previous papers), the proposed planner encourages such individuals to act pro-socially.

Deep reinforcement learning:

The authors used a graph neural network called GraphNet to conceptually encode the structure of the collaboration network. While the general approach has been clearly presented in the paper, I think more details are needed to explain the deep-learning part of the GraphNet-based method; this could help avoid any misunderstanding or confusion in the experimental section.

The authors should be more specific concerning the computational block configuration and the architecture they used to create the GN. For instance, they initially claimed that their model is a DL-based method, while they were presenting the parameters with a single layer MLP (as illustrated in Table S3). Is it possible to explain why RNNs neural nets have not been used to train the recommender NN, since this type is usually more suitable for time-dependent sequences, which is the case in this problem (especially if the contextual / mutual information of the cooperators was considered)?

Finally, it would be helpful to give more details of the used neural network's initialization and activation methods, and explain how fast is the GN converging (in terms of CPU time).

Potential extensions:

Even though the model is solid, I think one can significantly increase cooperation levels by adding simple instructions to the social-planner recommendations. In particular, the social planner could inform the players that, in the next iteration, it may advise other players to form social ties with positive reciprocators and cut ties with free riders and negative reciprocators. I believe such instruction would give individuals a chance to contemplate the possible consequences of any given action before they commit to that action, knowing that they may face the threat of exclusion from their social network.

Other comments:

An important aspect that is currently unclear is who pays for the central authority or the social planner? Moreover, if the cost of maintaining the social planner outweighs the benefits, then perhaps individuals would not opt for having such a recommendation system. I think these points are worth adding to the Discussion section.

In Figure 2, it might be interesting to compare the different solutions in terms of the distributions of payoffs that different players accumulate at the end of the game. Maybe the proposed social planner not only promotes cooperation, but also reduces payoff inequality.

The reader is repeatedly informed about the number of sessions in different conditions, but the authors never explain what a session means until page 9, when they clarify that it means a group of 16 players. This should be explained much earlier in the paper.

Reviewer #3:

Remarks to the Author:

"Scaffolding cooperation in human groups with deep reinforcement learning"

Using Deep Learning to develop a Social Planner, the authors experimentally showed that cooperation may be induced or elevated between humans by placing defectors in peripheral network positions, compared to three different control conditions. Given the general interest in cooperation and the exciting work on how networks/complex systems shape cooperation, this work should appeal to a broad audience. I have a major reservation about this work and several additional comments that I lay out below.

1. I believe the key result in this paper – higher cooperation rates – is driven by network density. How the graphs evolved was a black box in the main study, but in the validation study, that yielded oddly identical results (was the p-value really 1.00?), the probability of adding a tie between two cooperators was 1.00 (page 30). This resulted in very dense networks (page 29), including 2 maximally connected graphs between all cooperators and several with only few defectors.

The reason density is problematic as a solution here is that we know density is tied to cooperation in complex systems. Ohtsuki and colleagues explain in their abstract exactly why your results are unsurprising: "natural selection favors cooperation if the benefit of [cooperation], divided by the cost, exceed the average number of neighbors" (abstract). Your Planner, and its implementation in the validation study, gives cooperators more ties. Real world complex systems are not that dense, and as a result, more realistic tie formation processes need to be implemented. (Note also, this solves the paradox of defectors made more; cooperators were able to be exploited b/c of their number of ties to others)

Ohtsuki, H., Hauert, C., Lieberman, E., & Nowak, M. A. (2006). A simple rule for the evolution of cooperation on graphs and social networks. *Nature*, 441(7092), 502-505.

At a minimum, you need to address this work and how it is related to your results. I believe you should also be much more transparent about many of the results. Figure 4, for example, is one of the main results in the main text, and "influence" has nothing to do with what's graphed. Fig 4b is showing that giving cooperators the opportunity to have more ties results in them getting more ties ("the difference in average degree of cooperating participants and defecting participants"). It would be more informative to show network density by condition.

Specific comments:

2. In the abstract, the static network is a poor comparison for your Bot condition. The Rand-type dynamic network makes more sense, but it was constrained to not get as dense as your bot condition.

3. You probably do not mean to use "social capital."

4. I am aware of the existing work on how bots can intervene to increase cooperation, but I do not understand why we need that. You vaguely reference "social planners." Can you speak to the generalizability or external validity of all this? In what context could this be implemented?

5. First sentence of the 2nd paragraph. I don't believe there is any evidence that defectors sort with

other defectors. Much of this seems mischaracterized to me. What that work shows is preferential attachment to cooperators. This results in clustering, which is a network-level property that may be driven by preferential attachment, assortative mixing, and other relational mechanisms.

6. The first sign that something is “off” is Figure 2e. The cooperation rates are not that different, so why does the graph for your planner look so different? (around page 30 I think I came to the conclusion it's because the networks are so dense, they are cooperating with so many people)

7. When providing coefficients in the text, please refer readers to tables of coefficients. I wanted to look at your model specifications and was referred to graphs of marginal effects. I then had to go read the text corresponding to that, and so.

8. You have rounds nested in alters, alters nested in participants, and participants nested in networks. It seems you address two of these levels with the generalized mixed model. Really there is not a modeling framework for these data. We can address serial correlation and nesting, but the network effects cannot be addressed parametrically as far as I know. I think a discussion of these issues is warranted.

9. Towards the end of the results, “Rather than punish defectors with exclusion...” This was the first mention of network isolates. A key difference from your study and many prior others showing that dynamic networks increase cooperation is that your participants could not become isolated. In Rand et al 2011, Wang, Suri, and Watts 2012, and Melamed, Harrell, and Simpson 2018 (all in your references) network exclusion was possible. Defection is associated with network exclusion. Aside from the bot creating conditions with high density, another point of clarity is this issue. You should be clear about how your work is different than existing work – you don't allow isolates by design, while past work does.

10. Page 19. How did you decide the initial Game Parameters? E.g., the initial density is almost twice as dense as some past work.

11. Was this a between-subjects design? You don't specify (that I noticed), but past work has allowed participants to complete multiple conditions to increase the N.

Author Rebuttal to Initial comments

Point-by-point response for NATHUMBEHAV-22123206: “Scaffolding cooperation in human groups with deep reinforcement learning”

We deeply appreciate the effort and time our editor and reviewers spent providing feedback on our research. We have revised our manuscript and collected new empirical data to incorporate these suggestions. Below we provide a point-by-point response with specific references to the revised manuscript, giving particular attention to the two issues that the editor emphasized in their summary: Reviewer 3's first point (“over whether the results are trivially due to increased network density”) and

point 8 (on our statistical analysis). In the manuscript itself, we use red text to designate changes and revisions.

REVIEWER COMMENTS:

Reviewer #1:

Remarks to the Author:

This is a nifty paper on a profound topic, namely: how cooperation may be sustained within human groups, so as to avoid a tragedy of the commons that is so widely observed. It involves yet another challenging and important application of the sorts of artificial intelligence tools being developed at Deep Mind. The paper is just an initial foray into this topic (one that is, at least in this instantiation, admittedly less challenging, obviously, than AlphaGo or protein folding); but the paper is likely to push further an emerging body of work on the uses of AI to modify human group behavior. As such, it is likely to be of great interest to the readers of Nature Human Behavior. And the paper is well written and well executed.

My mostly small comments below are directed at clarifying some of the analysis and arguments.

We thank the reviewer for their thoughtful summary, and for the constructive feedback that they have provided in their comments.

I think it would be a bit clearer if the abstract (and rest of the paper) used the word “groups” instead of “sessions” to describe the various experiments.

We agree that this change would be clearer for readers. In the current revision, we replace “sessions” with “groups” throughout the paper, including in the abstract, results, and supplementary information.

At page 18, it is noted that the “social planner” (the AI agent) sits outside of and can observe the entirety of the social network as well as the cooperation decisions and capital levels of all the players. I think the body of the manuscript (for instance at page 2) should make it a bit clearer that the AI agent here has such global knowledge about the whole network of people and all their actions. To be clear, I like the contrast noted explicitly in footnote 2 about the difference between a distributed embodied approach and a centralized non-embodied approach. But this reality prompts the following concern: how this could scale

with network group size? Is there any way to estimate how scalable this is? One of the reasons, I gather, that the Shirado iScience paper uses multiple decentralized bots is precisely to avoid the demanding requirement of such global surveillance.

We thank the reviewer for identifying this important point. Scale has long been a central theme in cooperation research (e.g., Boyd & Richerson, 1988). We agree it remains a key challenge for modern research on mechanisms supporting cooperation.

Fortunately, scalability is also a major goal in the graph neural network field. Over the past five years, research on graph neural networks has directed a fair amount of attention to so-called “web-scale problems”, which involve “millions of nodes, billions of edges, and/or millions of graphs” (Addanki et al., 2021). Ying et al. (2018) provide an early example with their application of graph neural networks to construct “web-scale recommender systems” for Pinterest. Another good example is the work by Derrow-Pinion et al. (2021), deploying graph neural networks to significantly and globally improve time of arrival estimates for Google Maps users.

In the revised manuscript, we have added a corresponding explanation to the discussion (p. 8, lines 212-220):

Developments in machine learning indicate several promising directions for future research. The design of graph neural networks allows them to generalize to large-scale problems. Several teams, for example, have applied graph neural networks to so-called “web-scale” challenges, involving millions of nodes and potentially billions of edges (Hu et al., 2021; Ying et al., 2018). These successes hint at a path to scaffolding cooperation in expansive networks. Can an encouraging approach support community cohesion at large scales?

Another subtlety that I think needs to be noted (for instance at page 3) is how many times steps backwards the planner observes the “players’ recent decisions.” If it is the entire history of the players, this is also another demanding feature. At page 5, for example, it is made quite explicit that the planner changes its behavior as the game progresses – which highlights the temporality concerns again. Just like the issue of how the challenge would scale with increasing group size, one has to wonder how the AI planner would scale with increasing time steps and duration of interactions?

We appreciate the reviewer's point here, and have clarified the text accordingly. The GraphNet planner only sees players' decisions from the most recent round, and does not observe any decisions from prior rounds. Its planning strategy is thus agnostic to group member's prior decisions; in a direct sense, only their most recent choice matters. This limited observation allows us to construct a simple rule for the encouragement planner (see Tables S5-S7). We further discuss this design choice in response to Reviewer 2's question on recurrence.

We have added the following text to the main text (p. 2, lines 43-44, revised text in italics):

Every turn, the social planner observes the graph structure and the players' most recent decisions (i.e., their choice to cooperate or defect in the previous round).

We have also added a similar detail to the caption for Figure 1 (p. 2):

In this cooperation game, social planners observe the entire network of players and their cooperation decisions from the most recent round.

The change over rounds in the GraphNet planner's strategy that the reviewer mentions is actually an emergent property: social planners are not explicitly "informed" about time (i.e., round number is not included in their observations), and by design the GraphNet planner does not have a memory (i.e., it is "non-recurrent"). Nonetheless, the GraphNet planner learned a strategy that changes over time, potentially by encoding information through the graph changes it recommends. In the supporting information, we reference this detail as follows (p. 24, lines 648-649):

Despite the lack of recurrence, the agent could learn a time-conditional policy—for example, by decoding time information from player capital levels or by encoding time information in the structure of the graph itself.

I don't necessarily have a problem with the planner being trained on simulated human groups, especially given the sophistication of this Deep Mind team, but I couldn't help but wonder why the planner was not trained on real human groups?

The reviewer poses a great question here. In fact, it reflects a central research challenge within modern AI research.

Deep reinforcement learning can train neural networks to play games, molding any single network into an incredibly strong player. However, this network's strategy will only improve over large, large numbers of games (e.g., often in the millions, tens of millions, or more). A related common criticism of deep reinforcement learning is how inefficient it is relative to human learning (e.g., Marcus, 2018).

Typically an agent is trained with "self play," where it learns through games against copies of itself. Unfortunately, in coordination and mixed-motive games, training agents with self-play offers no guarantees that they will be compatible with humans (Carroll et al., 2019). An intuitive approach to improving human compatibility would be to train with humans ("human-in-the-loop reinforcement learning"). However, this imposes high labor costs and efficiency constraints on reinforcement learning: if the goal is to train an agent on a million rounds of a game and one human can play 100 rounds of a game with an agent per workday, it would take a week and a half with 1,000 humans working full time to finish training one copy of the agent. Of course, reinforcement learning experiments often require more than one million training rounds, and frequently involve many rounds of iteration or replication.

As eloquently summarized by Christiano et al. (2019), "In principle [training with humans] fits within the paradigm of reinforcement learning, but using human feedback directly as a reward function is prohibitively expensive for RL systems that require hundreds or thousands of hours of experience. In order to practically train deep RL systems with human feedback, we need to decrease the amount of feedback required by several orders of magnitude."

Simulation is one of the most common solutions to this challenge. Simulations can generate vast amounts of training rounds for reinforcement learning. As long as the simulated interaction partner behaves in a sufficiently human-like manner, the resulting policy should generalize to interacting with humans. (If the reviewer is familiar with ChatGPT, its training involves simulation of this sort. ChatGPT and other similar language models are trained with "reinforcement learning from human feedback" [RLHF], which simulates human feedback in order to support training efficiency.)

We have added a paragraph to the introduction to summarize this explanation (p. 3, lines 56-59):

Neural networks can learn through interaction with real human groups, but the amount of trial-and-error experience needed for deep reinforcement learning takes a generally prohibitive amount of time to accumulate. Interactions with simulated human groups enable our social planning agent to gain a large amount of experience in a short period of time.

For what it's worth, there are some important technical details regarding the implementation of the "cooperative clustering" algorithm (at page 20) that are different between this paper and the Shirado iScience paper. But these differences really don't matter for the argument this paper is making.

We appreciate this pointer. We attempted to summarize some of the implementation differences between Shirado and Christakis (2020) and our paper in Section S4.1 of the supporting information (p. 21), and would be very open to noting any additional differences.

I really like the observation about how the planner achieves core-periphery structure rather than simply clustering the defectors together on the edge of the graph (as noted on pages 7 and 8).

As I understand the modeling approach, the P values reported on page 6 and elsewhere arise from comparing individuals, not groups. That is, the sample size is the total number of people, not the total number of groups. This should be clarified. Also, does this statistical model described at the bottom of page 20 include group-level fixed effects?

We thank the reviewer for this question. They are correct that the cooperation model producing the "main" results reported on pp. 3-4 (lines 85-96) analyzes decisions at the individual level. We constructed the individual-level model following the example of several prior papers on this cooperation task (see Rand, Arbesman, & Christakis, 2011; Rand et al., 2014; Shirado et al., 2013; Shirado & Christakis, 2020). For example, Shirado and Christakis (2020) explain that they analyze behavior "at the level of individual cooperation decisions using a generalized linear mixed model (GLMM) with random effects for sessions with nested individuals" (p. 5).

All other models reported in the main text compare outcomes at the group level (e.g., assortative mixing), specifically in the final round of the game. Since each group contributes a single observation to the analysis, we are able to employ linear models for these analyses.

We have clarified these design choices and results with a footnote in the main text (p. 3):

Following past studies (Rand, Arbesman, & Christakis, 2011; Rand et al., 2014; Shirado et al., 2013; Shirado & Christakis, 2020), we employ generalized linear mixed models to analyze cooperation decisions at the individual level, with random effects for participants nested in groups. Linear models of cooperation at the group level echo the results from these individual-level models (results reported in Section S6.2 in supporting information). For all other game outcomes analyzed in the main text, we implement group-level linear models. Detailed model specifications are provided in the supporting information and within our analysis scripts available at <https://osf.io/8ahkg/>.

The statistical model described to page 9 has both the current rate of cooperation and the number of cooperating neighbors in the same model. I can understand why this parameterization was chosen, but surely those two variables are extremely collinear? At least that should be discussed?

This is a great question. The variance inflation factor (VIF) is often used to understand and diagnose multicollinearity issues. For this model, we estimate the VIF for the number of neighbors as 3.2, for the number of cooperating neighbors as 5.9, and for the rate of neighborhood cooperation as 3.5. A common rule of thumb is that VIF exceeding a threshold of 10 indicates substantial multicollinearity issues (Hadi & Chatterjee, 2012). By this standard, multicollinearity is not a pressing concern for the model.

Nonetheless, multiple statisticians and researchers argue against an overreliance on VIF as a diagnostic for multicollinearity. For example, O'Brien (2007) states that "threshold values of the VIF (and tolerance) need to be evaluated in the context of several other factors that influence the variance of regression coefficients" (p. 673), including sample size and the variance in the outcome explained by the predictor. Similarly, Fox (2016) explains that "the standard errors of the regression coefficients are the 'bottom line': If the coefficient estimates are sufficiently precise, then the degree of collinearity is irrelevant" (p. 341). With this in mind, we double checked to confirm that the confidence intervals for our model (depicted in Figure S4a) indicate precise effect estimates. Overall, we find this evidence reassuring with regards to multicollinearity.

In the revised text, we have added a new note about multicollinearity to the supporting information (p. 21, lines 601-603):

To test potential multicollinearity concerns, we compute the variance inflation factor (VIF) and inspect the precision of each effect estimate (Fox, 2015; O'Brien, 2007). The VIF for each fixed effect falls below standard rules of thumb, and the effect estimates are sufficiently precise that multicollinearity is not a substantial concern for the model.

The network group structure parameters in table S1 are standard and appropriate.

We thank the reviewer for checking our study parameters.

A bit more information on how nodes (subjects) that drop out are handled would be appreciated in the SI. I suspect that the investigators do not replace the dropouts with bots nor do they rewire the network around the dropped players. Probably they have the dropped player simply continue to play their last play (i.e., cooperation or defection)? But then what do the players do when offered the opportunity to make or cut ties by the planner? Maybe this was mentioned somewhere and I missed it?

We thank the reviewer for identifying this oversight, and appreciate the prompt to provide this information. As the reviewer suspected, we tried to be minimally disruptive to group dynamics when participants dropped out, and did not rewire the network to remove their nodes. Instead, we used a simple imitation rule that mimicked the decision making of participants in the focal group during the prior round. We excluded these imitation decisions from analysis.

We have added more information about our dropout procedure to the supplementary information (p. 20, lines 547-553):

Participants who dropped out were replaced with simple imitation bots for the remainder of the game. In each round, these imitation bots copied the overall decision making pattern of the participants in their group from the prior round. For example, if a bot replaced a participant after a round where 12 participants cooperated and three defected, in the next round it would cooperate with a 80% probability. Similarly, if the participants in its group accepted 75% of the recommendations they received in the previous round, it would accept each of its incoming recommendations with a 75% probability. These imitation decisions were excluded from analysis. We observed a very low dropout rate among our participants: groups completed the final round with a mean of 14.6 participants (median = 15) out of 16 still connected.

Another paper the authors might possibly wish to take a look at is this 2012 paper by Apicella et al. on cooperative clusters in Hadza hunter gathers: <https://www.nature.com/articles/nature10736>

We agree that this paper offers relevant insights for our manuscript, and thank the reviewer for sharing it. In the new revision, we have added a sentence to the introduction linking to the findings from both this paper and the related study by Smith et al. (2018; p. 1, lines 28-29):

Indeed, studies of modern hunter-gatherer tribes indicate that cooperative assortment may trace back to early epochs of human evolutionary history (Apicella et al., 2012; Smith et al., 2018).

Reviewer #2:

Remarks to the Author:

Summary of findings.

The paper addresses an important open question: how to encourage group cooperation and mitigate the spread of defection across the population. In their model, the authors present a social planner, which makes recommendations to rewire their ties with others in the network based on player's past actions as well as the network activity.

Fit to Nature Human Behaviour:

This research contribution is similar to the ones made by Shirado and Christakis, which got published in venues that are comparable Nature Human Behaviour (NHB). Moreover, the paper is generally well written, easy to follow, and addresses topics of broad appeal, making the paper suitable to a general audience. Hence, I believe the papers fits the scope of NHB.

Novelty:

While the paper does not make a theoretical contribution towards our understanding of human behavior, nor identifies a novel mechanism to establish cooperation, it does make a significant contribution towards understanding the potential use of AI to help solve social dilemmas. The authors neatly combine existing research to propose a solution, which uses Artificial Intelligence (AI) to facilitate cooperation, building on a similar line of research by Rand et al. [2011] as well as Shirado and Christakis [2020]. Importantly, unlike previous solutions, the authors' social planner is able to increase cooperation levels (rather than reduce the decline in cooperation levels) over the course of the game. I liked how the social planner learns a strategy from scratch, through repeated trial and error. I also like how, instead of ostracizing defectors (as is the case with previous papers), the proposed planner encourages such individuals to act pro-socially.

We thank the reviewer for their thoughtful and engaged review, including their comments on fit and novelty, and for the insightful suggestions they offer throughout their comments.

Deep reinforcement learning:

The authors used a graph neural network called GraphNet to conceptually encode the structure of the collaboration network. While the general approach has been clearly presented in the paper, I think more details are needed to explain the deep-learning part of the GraphNet-based method; this could help avoid any misunderstanding or confusion in the experimental section.

The authors should be more specific concerning the computational block configuration and the architecture they used to create the GN. For instance, they initially claimed that their model is a DL-based method, while they were presenting the parameters with a single layer MLP (as illustrated in Table S3).

We thank the reviewer for flagging this. As this comment highlights, each computational unit in the GraphNet module is shallow (a single layer). However, computations in a GraphNet module proceed in stages, starting with the edge-update function and followed by the node- and global-update functions. At a system level, there are at least three layers of computations between input and output in a GraphNet module. Moreover, the architecture for our social-planner agent connects two GraphNet modules in sequence, thus introducing a total of six layers between the input to the agent and its output. While the individual units are shallow, the overall architecture is deep.

We have expanded our discussion of our network architecture in the supporting information to incorporate these details (p. 26, lines 689-691):

Computations in a GraphNet module proceed in stages, passing through a sequence of update functions (see Section S5). The overall architecture comprises a total of six layers between the input to the agent and its output. Thus, while the individual computational units are shallow, the resulting architecture is deep.

Is it possible to explain why RNNs neural nets have not been used to train the recommender NN, since this type is usually more suitable for time-dependent sequences, which is the case in this problem (especially if the contextual / mutual information of the cooperators was considered)?

This is a great question. We designed our social-planner agent with a non-recurrent architecture to make it easier to interpret its policy. In particular, the analysis summarized in Figure 3 would be substantially more convoluted if the GraphNet planner was recurrent. In that situation, the planner's behavior would depend not only on its current observation, but also on the entire trajectory of the game up to that round.

We briefly experimented with a recurrent architecture and anecdotally found its performance to be no better than that of the purely feedforward agent. Given this initial experience and our interest in interpretability, we opted to focus on the feedforward architecture for our experiments.

We have added a summary of this discussion to the supporting information (p. 24, lines 647-651; new text in italics):

We designed this agent with a purely feedforward architecture to simplify the process of interpreting its policy. Despite the lack of recurrence, the agent could learn a time-conditional policy by decoding time information from player capital levels or by encoding time information in the structure of the graph itself. Nevertheless, the agent's policy and value estimate cannot directly depend on the graph history: all information must be deduced from the current state of the system.

Finally, it would be helpful to give more details of the used neural network's initialization and activation methods, and explain how fast is the GN converging (in terms of CPU time).

We thank the reviewer for flagging this omission. We have added a new table to the supporting information to provide further detail on the GraphNet architecture and parameterization (see Table S3, p. 26).

Computational unit	Architecture	Output size	Activation function	Initialization Function
Edge-update function ϕ_e^1	Single-layer MLP	128	tanh	Truncated normal
Node-update function ϕ_v^1	Single-layer MLP	128	tanh	Truncated normal
Global-update function ϕ_u^1	Single-layer MLP	128	tanh	Truncated normal
Edge-update function ϕ_e^2	Single-layer MLP	2	–	Truncated normal
Node-update function ϕ_v^2	Single-layer MLP	128	tanh	Truncated normal
Global-update function ϕ_u^2	Single-layer MLP	1	–	Truncated normal

We have also added the following textual explanation (p. 25, lines 686-688):

Each update function $\phi_{\{e,v,u\}}^{(1,2)}$ was instantiated through a multilayer perceptron (MLP) with a single layer. We use truncated normal initializers parameterized with $\mu = 0$ and $\sigma = 1 / \sqrt{\text{input size}}$ for the MLPs, truncating values at two standard deviations.

We likewise appreciate the request for the manuscript to provide details on the agent’s training time. We used accelerators to train our agent with one GPU per replicate, making CPU time difficult to compute. Wall time (which includes both compute time and potential delays) for the training experiment was

64,934 seconds, or roughly 18 hours. We have added the following text to section S5.5 (p. 25, lines 685-686):

Each replicate trains on one GPU, over approximately 18 hours of wall time.

Potential extensions:

Even though the model is solid, I think one can significantly increase cooperation levels by adding simple instructions to the social-planner recommendations. In particular, the social planner could inform the players that, in the next iteration, it may advise other players to form social ties with positive reciprocators and cut ties with free riders and negative reciprocators. I believe such instruction would give individuals a chance to contemplate the possible consequences of any given action before they commit to that action, knowing that they may face the threat of exclusion from their social network.

This is a great idea, especially given the advent and proliferation of large language models (LLMs; Brown et al., 2020; Devlin et al., 2019). Recent explorations in language research have applied LLMs to offer explanations for algorithmic decision making (e.g., Wiegrefe et al., 2022). We have added the following comment to the discussion (p. 8, lines 216-220):

Another potential path concerns interpretability. Recent work demonstrates that large language models (e.g., Brown et al., 2020; Devlin et al., 2019) may be capable of generating explanations for algorithmic decision making (Wiegrefe et al., 2022). With the support of a language model, our social planner could explain its policy to group members in natural language, helping them to understand the possible consequences of any choices that they might make.

Other comments:

An important aspect that is currently unclear is who pays for the central authority or the social planner? Moreover, if the cost of maintaining the social planner outweighs the benefits, then perhaps individuals would not opt for having such a recommendation system. I think these points are worth adding to the Discussion section.

We agree that this is an important question, and have added the following text to the discussion (p. 8, lines 228-232):

The deployment of agents to assist with social planning raises related questions concerning consent and governance. Which stakeholders should direct, steer, and fund AI systems in this domain? The application of participatory and democratic methods will be particularly important for such technology (Birhane et al., 2022; Garvey et al., 2018). It is imperative that technologists preserve the ability of communities that will be affected by AI to engage with it on their own terms—whether that is to withdraw from, contribute to, steer, or potentially resist the deployment of these systems.

In Figure 2, it might be interesting to compare the different solutions in terms of the distributions of payoffs that different players accumulate at the end of the game. Maybe the proposed social planner not only promotes cooperation, but also reduces payoff inequality.

This is another great suggestion. We have added a new row to Figure 2, now labeled Figure 2f, that shows Lorenz curves for the final round of each group in each condition. Lorenz curves are a popular visual representation of group inequality (Lorenz, 1905; see also Gastwirth, 1972). In these plots, the 45° line represents perfect equality; the curves represent individual groups, with lower curves reflecting lower equality. The GraphNet planner substantially improves equality over the baselines, as these new plots indicate (p. 4; see also Figure S1d).

The reader is repeatedly informed about the number of sessions in different conditions, but the authors never explain what a session means until page 9, when they clarify that it means a group of 16 players. This should be explained much earlier in the paper.

We agree with the reviewer that this change would be clearer for readers. In the current revision, we replace “sessions” with “groups” throughout the paper, and clarify in the introduction that groups consist of 16 players (p. 3, lines 82-83).

Reviewer #3:

Remarks to the Author:

"Scaffolding cooperation in human groups with deep reinforcement learning"

Using Deep Learning to develop a Social Planner, the authors experimentally showed that cooperation may be induced or elevated between humans by placing defectors in peripheral network positions, compared to three different control conditions. Given the general interest in cooperation and the exciting work on how networks/complex systems shape cooperation, this work should appeal to a broad audience. I have a major reservation about this work and several additional comments that I lay out below.

We thank the reviewer for their succinct summary of our approach and design, and for the thoughtful feedback they offer.

1. I believe the key result in this paper – higher cooperation rates – is driven by network density. How the graphs evolved was a black box in the main study, but in the validation study, that yielded oddly identical results (was the p-value really 1.00?), the probability of adding a tie between two cooperators was 1.00 (page 30). This resulted in very dense networks (page 29), including 2 maximally connected graphs between all cooperators and several with only few defectors.

The reason density is problematic as a solution here is that we know density is tied to cooperation in complex systems. Ohtsuki and colleagues explain in their abstract exactly why your results are unsurprising: “natural selection favors cooperation if the benefit of [cooperation], divided by the cost, exceed the average number of neighbors” (abstract). Your Planner, and its implementation in the validation study, gives cooperators more ties. Real world complex systems are not that dense, and as a result, more realistic tie formation processes need to be implemented. (Note also, this solves the paradox of defectors made more; cooperators were able to be exploited b/c of their number of ties to others)

Ohtsuki, H., Hauert, C., Lieberman, E., & Nowak, M. A. (2006). A simple rule for the evolution of cooperation on graphs and social networks. *Nature*, 441(7092), 502-505.

At a minimum, you need to address this work and how it is related to your results.

We thank the reviewer for proposing this alternative hypothesis. To summarize the suggested hypothesis: network density is a proximal cause of cooperation in this group game, and thus high density *alone* explains the high levels of group cooperation that we observe in the GraphNet planner condition. This proposal prompted us to take a step back to re-examine our findings and those of Ohtsuki et al. (2012).

A close reading of Ohtsuki et al. (2012) offers initial evidence against the proposed alternative hypothesis. When the reviewer quotes the abstract from Ohtsuki et al. (2012), they reference the titular “simple rule”:

$$b/c > k$$

where b is the benefit of cooperation, c is the cost of cooperation, and k represents the average number of neighbors for all players in the network. Notably, b and c are static game parameters. For example, in our experiments, $b = 0.1$ and $c = 0.05$. Increasing density increases the average number of neighbors (k). Since the left-hand side of the inequality is fixed, this increase makes cooperation *less likely*, not more. To quote the authors’ interpretation of the simple rule directly from their paper, “certain network structures should promote cooperative behaviour more than others. In particular, more cooperation should emerge if connectivity is low [...] [and] higher connectivity should reduce cooperation” (p. 3).

Nonetheless, to directly test the alternative hypothesis that “the key result in this paper – higher cooperation rates – is driven by network density”, we designed and collected new data in two additional experimental conditions (total $N = 400$ across 25 groups).

The first of these two conditions evaluates a social planner that attempts to replicate the connectivity levels seen under the GraphNet planner, without conditioning on player choices. The strategy of this “neutral planner” represents an implicit assumption that the GraphNet planner’s “encouraging” approach is not necessary to generate high network connectivity and high cooperation levels. Second, we construct a social planner that solely attempts to maximize node connectedness. On every round, this “maximum connectivity” planner recommends that every unlinked pair of players in the graph establish a connection. If density is the proximal cause of cooperation, groups in these conditions should either match or exceed the cooperation levels observed in the GraphNet planner.

Results from these two conditions verify that higher density does not support greater cooperation. We report the results of these conditions in the main text (p. 7, lines 169-182; key results in bold):

The recommendations from both the GraphNet planner and the encouragement planner produce networks with notably high density, especially relative to the baseline conditions (see Figure S1b). Under the GraphNet planner, for example, several groups reached full connectivity (see Figure S11).

*To evaluate the possibility that high density alone drives the success of these planners—without the need for an encouraging approach—we run two additional follow-up studies ($N = 400$ participants across 25 groups). First, we build a “neutral” social planner that aims to recreate the connectivity dynamics observed under the GraphNet planner, without regard for player’s choices (i.e., dispensing with the encouraging approach to defectors; $N = 192$ participants across 12 groups). As intended, this planner generates levels of network connectivity to a similar extent as the GraphNet planner (linear model; $p = 0.47$). **Nonetheless, its choice-agnostic approach degrades group cooperation significantly over time (generalized linear mixed model; coeff = -0.17 , $p < 0.001$).** As a further test of whether network density drives the high cooperation rates seen with the GraphNet planner, we construct another social planner that seeks to maximize network connectivity as much as possible ($N = 208$ participants across 13 groups). **This strategy generates levels of network density that significantly exceed those produced by the GraphNet planner ($p < 0.001$).** However, this also causes a precipitous decline in cooperation (coeff = -0.51 , $p < 0.001$). On its own, network density does not offer a compelling explanation for the high cooperation rates supported by the GraphNet planner.*

These follow-up studies provide substantial evidence that high density alone does not explain the performance of our GraphNet social planner.

Additional minor comments:

- Concerning our analysis, the reviewer asks: “The validation study [...] yielded oddly identical results (was the p -value really 1.00?)”
 - At two decimal places, the p -value for this comparison is indeed 1.00. The results are available on OSF for closer inspection, at <https://osf.io/8ahkg/>.
- Concerning the design of the experimental game, the reviewer suggests: “Real world complex systems are not that dense, and as a result, more realistic tie formation processes need to be implemented.”
 - This domain and its connectivity rules are well defined by prior papers (e.g., Nishi et al., 2015; Rand, Arbesman, & Christakis, 2011; Shirado et al., 2013). We agree that translational research linking these laboratory-style experiments to the real world would be incredibly valuable, but believe such a study falls outside the scope of the current work.

- Concerning the relative payoff for defection in the GraphNet planner condition, the reviewer writes: “Note also, this solves the paradox of defectors made more; cooperators were able to be exploited b/c of their number of ties to others”.
 - With our analysis of Ohtuski et al. (2012), we are not sure that this resolves the issue. Their framework is quite clear that cooperation should not spread when cooperators are exploited over a large number of ties. The co-existence of the observed payoff gap with persistently high levels of cooperation is a surprise, given the strong claims prior papers make that cooperators must earn more for group cooperation to increase. For example, Rand, Arbesman, & Christakis (2011) write, “How can this high level of cooperation be maintained in the face of possible exploitation? One answer [...] [is that] cooperators are more likely to interact with other cooperators and therefore earn higher payoffs” (p. 19193). The combination of increasing cooperation levels and the relative payoff advantage for defection defies the rational, incentive-based choice frameworks that prior papers invoke.

I believe you should also be much more transparent about many of the results. Figure 4, for example, is one of the main results in the main text, and “influence” has nothing to do with what’s graphed. Fig 4b is showing that giving cooperators the opportunity to have more ties results in them getting more ties (“the difference in average degree of cooperating participants and defecting participants”). It would be more informative to show network density by condition.

As the reviewer suggests, we have changed our terminology throughout the paper, replacing “influence” with “relative degree” or “relative connectivity”. We agree that it is informative to see how network degree varies by condition, as depicted in Figure S1b.

Specific comments:

2. In the abstract, the static network is a poor comparison for your Bot condition. The Rand-type dynamic network makes more sense, but it was constrained to not get as dense as your bot condition.

We thank the reviewer for this comment, and respectfully disagree. The static network condition is a natural counterfactual for a social planner: what if, without any intervention at all, outcomes improve to a similar degree? Nonetheless, the revised abstract also includes a reference to the random-

recommendations condition, as well as a reference to the new conditions that directly address the reviewer's density hypothesis.

3. You probably do not mean to use “social capital.”

We agree with the reviewer, and replace the phrase “social capital” in the first paragraph with “amity”.

4. I am aware of the existing work on how bots can intervene to increase cooperation, but I do not understand why we need that. You vaguely reference “social planners.” Can you speak to the generalizability or external validity of all this? In what context could this be implemented?

We thank the reviewer for this question. The “social planner” concept originated in welfare economics, and has become a common term across different fields in the social sciences. In its precise, original usage, the term refers to an individual or institution who can make structural decisions and who seeks to maximize social welfare. Canonical references to the concept include *Intermediate Public Economics* (Hindricks & Miles, 2013) and “Equilibrium and Its Basic Welfare Properties” in *Microeconomic Theory* (Mas-Colell, Whinston, & Green, 1995). For an example in this journal, see Beckage, Moore, & Lacasse (2022). Over the years, the term has acquired a broader connotation across the social sciences. Thaler and Sunstein (2003), for example, define social planners as “anyone who must design plans for others, from human resource directors to bureaucrats to kings” (p. 178).

For our domain of network connectivity, social planning includes both mundane, in-person decisions and wide-scale, algorithmic decisions:

- In the former case, a person acts as a networking social planner any time that they make a connection between two other individuals. Imagine a party host who greets a late-arriving guest, and faces the choice of which group to direct them to join. Most guests arrive in good moods; some arrive in bad moods. In this example, the (intuitive) insight is—paraphrasing Reviewer 1—to avoid simply clustering the unhappy guests together on the edge of the party, and instead distribute them to happier clusters throughout the party.
- In the latter case, algorithms play a key role in contemporary social systems, acting as planners in online social networks. These planner algorithms mediate the structure of online networks by recommending connections between nodes. For example, algorithms recommend connections on

Twitter (“who to follow”), Instagram (“discover people”), LinkedIn (“more suggestions for you”), and other social media platforms. We discuss the role that algorithms play in online social networks in the main text on page 2 (lines 30-35).

We have also added a new explanation of the “social planner” term to the supporting information (p. 18):

In its precise, original usage, “social planner” refers to an individual or institution who can make structural decisions and who seeks to maximize social welfare (Hindricks & Miles, 2013; Mas-Colell, Whinston, & Green, 1995). Over the years, the term has acquired a broader connotation in the social sciences. Thaler and Sunstein (2003), for example, define social planners as “anyone who must design plans for others, from human resource directors to bureaucrats to kings” (p. 178).

5. First sentence of the 2nd paragraph. I don’t believe there is any evidence that defectors sort with other defectors. Much of this seems mischaracterized to me. What that work shows is preferential attachment to cooperators. This results in clustering, which is a network-level property that may be driven by preferential attachment, assortative mixing, and other relational mechanisms.

We thank the reviewer for this feedback. We agree that a large amount of work focuses on the tendency of cooperators to connect with other cooperators. In our survey of prior research, we additionally reviewed papers including defectors sorting with other defectors in their definition of clustering and assortativity. For example, Shirado and Christakis (2020) analyze choice assortativity over the study conditions they ran, measuring assortativity as the correlation between neighbors’ choices (see p. 3 of their supporting information, Figure S1d). Santos et al. (2006) evaluate the effect of strategy and structural updates on the local assortativity (and disassortativity) of player strategies. To study “assortativity in cooperation”, Apicella et al. (2012) measure the correlation between an individual’s cooperation choice and their neighbor’s cooperation choice, measured as the number of units one donated in a public goods game. In their investigation of strategy “clustering” and “assortment”, Rand et al. (2014) demonstrate the ways that both cooperators and defectors tend to cluster (see Figure 4b and 4c on p. 17095). Wang, Suri, and Watts (2012) build an explicit measure of defector-defector assortativity to analyze network outcomes for their studies.

Following the reviewer’s feedback and another review of our citations, we have removed the reference to Melamed, Harrell, and Simpson (2018) from this passage. Though their article title includes the phrase “assortative mixing”, the actual subject matter of the article focuses on structural clustering (vis-à-vis Watts & Strogatz, 1988) rather than strategy clustering (as in the other references).

If the reviewer can recommend a contrasting reference laying out the argument that they have in mind, we would be happy to consider adding it.

6. The first sign that something is “off” is Figure 2e. The cooperation rates are not that different, so why does the graph for your planner look so different? (around page 30 I think I came to the conclusion its because the networks are so dense, they are cooperating with so many people)

We thank the reviewer for this feedback. We agree that the networks in Figure 2e look visually different. This difference stems from the recommendation strategy that the GraphNet planner learns, which steers groups in particular directions, affecting several key graph-level outcomes such as density and core-periphery structure (see Figures S1b and S2d).

While the cooperation rates appear visually similar to the reviewer, multiple models indicate that the cooperation rates differ significantly. For example, through pairwise contrasts between estimated marginal means, the mixed-effects model regressing changes in cooperation on condition indicates significant differences between the GraphNet planner and the static network condition ($p < 0.001$), the random recommendations condition ($p < 0.001$), and the cooperative clustering condition ($p < 0.001$).

7. When providing coefficients in the text, please refer readers to tables of coefficients. I wanted to look at your model specifications and was referred to graphs of marginal effects. I then had to go read the text corresponding to that, and so.

We appreciate this recommendation. As with the prior draft, the revised supporting information provides model specifications and coefficients in Sections S6.2 and S7.3, in addition to plots visualizing marginal effects (Figures S8 and S12). We have included our analysis scripts and cleaned data (<https://osf.io/8ahkg/>) in this revision, which provide detailed model specifications and outputs. We find that these scripts provide statistically inclined readers with exact, transparent results, minimizing the chance of transcription errors and encouraging replication analysis, while also providing an easily interpretable textual description for more-casual readers.

8. You have rounds nested in alters, alters nested in participants, and participants nested in networks. It seems you address two of these levels with the generalized mixed model. Really there is not a modeling

framework for these data. We can address serial correlation and nesting, but the network effects cannot be addressed parametrically as far as I know. I think a discussion of these issues is warranted.

We thank the reviewer for the insightful question about our statistical modeling.

First, we note that alters are not relevant to the statistical analysis in question. To clarify a potential source of confusion, in some prior studies of networked cooperation, participants played a networked prisoner's dilemma game (e.g., Gracia-Lázaro et al., 2012; Harrell, Melamed, & Simpson, 2018). In this sort of dyadic domain, statistical models of participant cooperation account for alters because players make multiple choices per round (e.g., see p. 9 in the supporting information for Harrell, Melamed, & Simpson, 2018). In contrast, our participants play a networked game akin to a public goods game: rather than choosing one independent action per neighbor, players choose a single binary action per round that simultaneously applies to all of their neighbors. As a result, statistical models in the public goods-style domain do not need to account for alters, since players make a single choice per round.

Our “main” result, concerning changes in cooperation rates over time, is implemented as a generalized linear mixed model with a logistic link function. Following prior papers (Rand, Arbesman, & Christakis, 2011; Rand et al., 2014; Shirado et al., 2013; Shirado & Christakis, 2020), we account for the nested data structure with nested random effects (participants within groups). The model includes a common intercept and round number as a fixed effect (in interaction with condition). This approach estimates a common starting level of cooperation (the intercept), and a rate of change in cooperation over time for each condition (the round-condition fixed effects).

All other regressions in the original submission were implemented as linear models analyzing outcomes in the final round. The outcomes are computed at the network level—for example, assortative mixing or degree of core-periphery structure. They are thus computed once per round per group; because the models examine outcomes in the final round, each group provides only one observation to each model, with no overlap in either participants or groups in each model. These regressions thus avoid the issues (serial correlation and nesting) suggested by the reviewer.

We have also clarified these design choices and results with a footnote in the main text (p. 3):

Following past studies (Shirado et al., 2013; Shirado & Christakis, 2020), we employ generalized linear mixed models to analyze cooperation decisions at the individual level (for detailed model

specification, see supporting information). Group-level models of cooperation echo the results from these individual-level models (see Section S6.2 in supporting information). For all other game outcomes analyzed in the main text, we implement group-level linear models.

9. Towards the end of the results, “Rather than punish defectors with exclusion...” This was the first mention of network isolates. A key difference from your study and many prior others showing that dynamic networks increase cooperation is that your participants could not become isolated. In Rand et al 2011, Wang, Suri, and Watts 2012, and Melamed, Harrell, and Simpson 2018 (all in your references) network exclusion was possible. Defection is associated with network exclusion. Aside from the bot creating conditions with high density, another point of clarity is this issue. You should be clear about how your work is different than existing work – you don’t allow isolates by design, while past work does.

We appreciate the reviewer’s concern on this point. Our game design indeed allows for network isolates. We now explicitly emphasize this information in the introduction (p. 3, lines 46-48):

The game makes no constraints on graph structure aside from preventing self-loops: with the right circumstances and recommendations, a social planner can produce isolates or fully connected networks.

We reiterate this detail in the supporting information (p. 18, line 494):

Players may become fully isolated as a result of the accepted edge changes.

10. Page 19. How did you decide the initial Game Parameters? E.g., the initial density is almost twice as dense as some past work.

We appreciate this feedback. We chose the initial game parameters by drawing from recent work on this domain. For example, Shirado and Christakis (2020) organized participants into networks “with an Erdős-Rényi random graph configuration in which 30% of the possible ties were present at the outset, on average” (p. 2). Nishi et al. (2015) similarly initialize networks “with an Erdős-Rényi random graph configuration in which 30% of ties were present” (p. 426). We note that Reviewer 1 confirms that “the network group structure parameters in table S1 are standard and appropriate.”

11. Was this a between-subjects design? You don't specify (that I noticed), but past work has allowed participants to complete multiple conditions to increase the N.

We thank the reviewer for this question. This was a between-participants design: participants completed a single condition. We have included the following text in the methods section (p. 9, lines 266-267):

The experiments employed a between-participants design: that is, participants joined a single group (with no participant experiencing multiple conditions).

References

- Addanki, R., Battaglia, P. W., Budden, D., Deac, A., Godwin, J., Keck, T., ... & Veličković, P. (2021). Large-scale graph representation learning with very deep GNNs and self-supervision. *Open Graph Benchmark Large-Scale Challenge*.
- Apicella, C. L., Marlowe, F. W., Fowler, J. H., & Christakis, N. A. (2012). Social networks and cooperation in hunter-gatherers. *Nature*, *481*(7382), 497-501.
- Beckage, B., Moore, F. C., & Lacasse, K. (2022). Incorporating human behaviour into Earth system modelling. *Nature Human Behaviour*, 1-10.
- Birhane, A., Isaac, W., Prabhakaran, V., Díaz, M., Elish, M. C., Gabriel, I., & Mohamed, S. (2022). Power to the people? Opportunities and challenges for participatory AI. *Equity and Access in Algorithms, Mechanisms, and Optimization*, 1-8.
- Boyd, R., & Richerson, P. J. (1988). The evolution of reciprocity in sizable groups. *Journal of Theoretical Biology*, *132*(3), 337-356.
- Brown, T., Mann, B., Ryder, N., Subbiah, M., Kaplan, J. D., Dhariwal, P., ... & Amodei, D. (2020). Language models are few-shot learners. *Advances in Neural Information Processing Systems*, *33*, 1877-1901.
- Carroll, M., Shah, R., Ho, M. K., Griffiths, T., Seshia, S., Abbeel, P., & Dragan, A. (2019). On the utility of learning about humans for human-AI coordination. *Advances in Neural Information Processing Systems*, *32*, 1-12.
- Christiano, P. F., Leike, J., Brown, T., Martic, M., Legg, S., & Amodei, D. (2017). Deep reinforcement learning from human preferences. *Advances in Neural Information Processing Systems*, *30*.

- Derrow-Pinion, A., She, J., Wong, D., Lange, O., Hester, T., Perez, L., ... & Velickovic, P. (2021, October). ETA prediction with graph neural networks in Google Maps. In *Proceedings of the 30th ACM International Conference on Information & Knowledge Management* (pp. 3767-3776).
- Devlin, J., Chang, M. W., Lee, K., & Toutanova, K. (2019, June). BERT: Pre-training of deep bidirectional transformers for language understanding. In *Proceedings of the 2019 Conference of the North American Chapter of the Association for Computational Linguistics: Human Language Technologies* (pp. 4171-4186).
- Fox, J. (2015). *Applied regression analysis and generalized linear models*. Sage Publications.
- Garvey, C. (2018, April). A framework for evaluating barriers to the democratization of artificial intelligence. In *Proceedings of the AAAI conference on Artificial Intelligence* (pp. 8079-8080).
- Gastwirth, J. L. (1972). The estimation of the Lorenz curve and Gini index. *The Review of Economics and Statistics*, 306-316.
- Gracia-Lázaro, C., Ferrer, A., Ruiz, G., Tarancón, A., Cuesta, J. A., Sánchez, A., & Moreno, Y. (2012). Heterogeneous networks do not promote cooperation when humans play a prisoner's dilemma. *Proceedings of the National Academy of Sciences*, 109(32), 12922-12926.
- Hadi, A. S., & Chatterjee, S. (2015). *Regression analysis by example*. John Wiley & Sons.
- Harrell, A., Melamed, D., & Simpson, B. (2018). The strength of dynamic ties: The ability to alter some ties promotes cooperation in those that cannot be altered. *Science Advances*, 4(12), eaau9109.
- Hindriks, J., & Myles, G. D. (2013). *Intermediate public economics*. MIT Press.
- Hu, W., Fey, M., Ren, H., Nakata, M., Dong, Y., & Leskovec, J. (2021, December). OGB-LSC: A Large-Scale Challenge for Machine Learning on Graphs. In *Proceedings of the Thirty-Fifth Conference on Neural Information Processing Systems Datasets and Benchmarks Track* (pp. 1-15).
- Lorenz, M. O. (1905). Methods of measuring the concentration of wealth. *Publications of the American Statistical Association*, 9(70), 209-219.
- Marcus, G. (2018). Deep learning: A critical appraisal. *arXiv preprint arXiv:1801.00631*.
- Mas-Colell, A., Whinston, M. D., & Green, J. R. (1995). *Microeconomic theory*. Oxford University Press.
- Melamed, D., Harrell, A., & Simpson, B. (2018). Cooperation, clustering, and assortative mixing in dynamic networks. *Proceedings of the National Academy of Sciences*, 115(5), 951-956.
- Nishi, A., Shirado, H., Rand, D. G., & Christakis, N. A. (2015). Inequality and visibility of wealth in experimental social networks. *Nature*, 526(7573), 426-429.

- O'Brien, R. M. (2007). A caution regarding rules of thumb for variance inflation factors. *Quality & Quantity*, *41*, 673-690.
- Ohtsuki, H., Hauert, C., Lieberman, E., & Nowak, M. A. (2006). A simple rule for the evolution of cooperation on graphs and social networks. *Nature*, *441*(7092), 502-505.
- Rand, D. G., Arbesman, S., & Christakis, N. A. (2011). Dynamic social networks promote cooperation in experiments with humans. *Proceedings of the National Academy of Sciences*, *108*(48), 19193-19198.
- Rand, D. G., Nowak, M. A., Fowler, J. H., & Christakis, N. A. (2014). Static network structure can stabilize human cooperation. *Proceedings of the National Academy of Sciences*, *111*(48), 17093-17098.
- Santos, F. C., Pacheco, J. M., & Lenaerts, T. (2006). Cooperation prevails when individuals adjust their social ties. *PLoS Computational Biology*, *2*(10), e140.
- Shirado, H., & Christakis, N. A. (2020). Network engineering using autonomous agents increases cooperation in human groups. *iScience*, *23*(9), 101438.
- Shirado, H., Fu, F., Fowler, J. H., & Christakis, N. A. (2013). Quality versus quantity of social ties in experimental cooperative networks. *Nature Communications*, *4*(1), 2814.
- Smith, K. M., Larroucau, T., Mabulla, I. A., & Apicella, C. L. (2018). Hunter-gatherers maintain assortativity in cooperation despite high levels of residential change and mixing. *Current Biology*, *28*(19), 3152-3157.
- Thaler, R. H., & Sunstein, C. R. (2003). Libertarian paternalism. *American Economic Review*, *93*(2), 175-179.
- Strouse, D., McKee, K. R., Botvinick, M., Hughes, E., & Everett, R. (2021). Collaborating with humans without human data. *Advances in Neural Information Processing Systems*, *34*, 14502-14515.
- Wang, J., Suri, S., & Watts, D. J. (2012). Cooperation and assortativity with dynamic partner updating. *Proceedings of the National Academy of Sciences*, *109*(36), 14363-14368.
- Watts, D. J., & Strogatz, S. H. (1998). Collective dynamics of 'small-world' networks. *Nature*, *393*(6684), 440-442.
- Wiegrefe, S., Hessel, J., Swayamdipta, S., Riedl, M., & Choi, Y. (2022, July). Reframing human-AI collaboration for generating free-text explanations. In *Proceedings of the 2022 Conference of the North American Chapter of the Association for Computational Linguistics: Human Language Technologies* (pp. 632-658).

Ying, R., He, R., Chen, K., Eksombatchai, P., Hamilton, W. L., & Leskovec, J. (2018, July). Graph convolutional neural networks for web-scale recommender systems. In *Proceedings of the 24th ACM SIGKDD International Conference on Knowledge Discovery & Data Mining* (pp. 974-983).

Decision Letter, first revision:

26th June 2023

Dear Dr. McKee,

Thank you for your patience as we've prepared the guidelines for final submission of your Nature Human Behaviour manuscript, "Scaffolding cooperation in human groups with deep reinforcement learning" (NATHUMBEHAV-22123206A). Please carefully follow the step-by-step instructions provided in the attached file, and add a response in each row of the table to indicate the changes that you have made. Please also address the additional marked-up edits we have proposed within the reporting summary. Ensuring that each point is addressed will help to ensure that your revised manuscript can be swiftly handed over to our production team.

We would hope to receive your revised paper, with all of the requested files and forms within two-three weeks. Please get in contact with us if you anticipate delays.

If you have not done so already, please alert us to any related manuscripts from your group that are under consideration or in press at other journals, or are being written up for submission to other journals (see:

<https://www.nature.com/nature-research/editorial-policies/plagiarism#policy-on-duplicate-publication> for details).

Nature Human Behaviour offers a Transparent Peer Review option for new original research manuscripts submitted after December 1st, 2019. As part of this initiative, we encourage our authors to support increased transparency into the peer review process by agreeing to have the reviewer comments, author rebuttal letters, and editorial decision letters published as a Supplementary item. When you submit your final files please clearly state in your cover letter whether or not you would like to participate in this initiative. Please note that failure to state your preference will result in delays in accepting your manuscript for publication.

In recognition of the time and expertise our reviewers provide to Nature Human Behaviour's editorial process, we would like to formally acknowledge their contribution to the external peer review of your manuscript entitled "Scaffolding cooperation in human groups with deep reinforcement learning". For those reviewers who give their assent, we will be publishing their names alongside the published article.

Cover suggestions

As you prepare your final files we encourage you to consider whether you have any images or illustrations that may be appropriate for use on the cover of Nature Human Behaviour.

ORCID

Non-corresponding authors do not have to link their ORCIDs but are encouraged to do so. Please note that it will not be possible to add/modify ORCIDs at proof. Thus, please let your co-authors know that if they wish to have their ORCID added to the paper they must follow the procedure described in the following link prior to acceptance:

Nature Human Behaviour has now transitioned to a unified Rights Collection system which will allow our Author Services team to quickly and easily collect the rights and permissions required to publish your work. Approximately 10 days after your paper is formally accepted, you will receive an email in providing you with a link to complete the grant of rights. If your paper is eligible for Open Access, our Author Services team will also be in touch regarding any additional information that may be required to arrange payment for your article. Please note that you will not receive your proofs until the publishing

agreement has been received through our system.

Please note that *Nature Human Behaviour* is a Transformative Journal (TJ). Authors may publish their research with us through the traditional subscription access route or make their paper immediately open access through payment of an article-processing charge (APC). Authors will not be required to make a final decision about access to their article until it has been accepted. Find out more about Transformative Journals

[REDACTED]

Best regards,
Alex McKay
Editorial Assistant
Nature Human Behaviour

On behalf of

Jamie

Dr Jamie Horder
Senior Editor
Nature Human Behaviour

Reviewer #1:

Remarks to the Author:

I think the authors did a very fine job of revision in response to all three reviews, and I have no further comments.

Reviewer #2:

Remarks to the Author:

The authors have addressed all of my concerns, and I am satisfied with the authors comments. Based on this, I recommend the paper for publication at Nature Human Behaviour

Final Decision Letter:

Dear Mr McKee,

We are pleased to inform you that your Article "Scaffolding cooperation in human groups with deep reinforcement learning", has now been accepted for publication in Nature Human Behaviour.

Please note that *Nature Human Behaviour* is a Transformative Journal (TJ). Authors whose manuscript was submitted on or after January 1st, 2021, may publish their research with us through the traditional subscription access route or make their paper immediately open access through payment of an article-processing charge (APC). Authors will not be required to make a final decision about access to their article until it has been accepted. IMPORTANT NOTE: Articles submitted before January 1st, 2021, are not eligible for Open Access publication. Find out more about Transformative Journals

With best regards,

Jamie

Dr Jamie Horder
Senior Editor
Nature Human Behaviour